# Chromosome end protection by RAP1-mediated inhibition of DNA-PK

Patrik Eickhoff[1], Ceylan Sonmez[2], Charlotte E. L. Fisher[1], Oviya Inian[3], Theodoros I. Roumeliotis[4], Angela dello Stritto[2], Jörg Mansfeld[5], Jyoti S. Choudhary[4], Sebastian Guettler[3], Francisca Lottersberger[2 ✉] & Max E. Douglas[1 ✉]

During classical non-homologous end joining (cNHEJ), DNA-dependent protein kinase (DNA-PK) encapsulates free DNA ends, forming a recruitment platform for downstream end-joining factors including ligase 4 (LIG4)[1]. DNA-PK can also bind telomeres and regulate their resection[2–4], but does not initiate cNHEJ at this position. How the end-joining process is regulated in this context-specific manner is currently unclear. Here we show that the shelterin components TRF2 and RAP1 form a complex with DNA-PK that directly represses its end-joining function at telomeres. Biochemical experiments and cryo-electron microscopy reveal that when bound to TRF2, RAP1 establishes a network of interactions with KU and DNA that prevents DNA-PK from recruiting LIG4. In mouse and human cells, RAP1 is redundant with the Apollo nuclease in repressing cNHEJ at chromosome ends, demonstrating that the inhibition of DNA-PK prevents telomere fusions in parallel with overhang-dependent mechanisms. Our experiments show that the end-joining function of DNA-PK is directly and specifically repressed at telomeres, establishing a molecular mechanism for how individual linear chromosomes are maintained in mammalian cells.

Mammalian telomeres are protected from cNHEJ by the shelterin subunit TRF2, which is proposed to hide the chromosome end from DNA repair factors by forming a lariat structure referred to as a t-loop[2,5,6]. T-loop assembly requires a terminal 3′ overhang[7], which at blunt telomeres resulting from leading strand DNA replication is formed via a 5′ resection step mediated by Apollo exonuclease[8–12]. DNA-PK is required for this resection step, but is unable to activate cNHEJ even after Apollo deletion[13], suggesting that a mechanism is in place to directly block the end-joining process at telomeres. We hypothesized that this mechanism may involve the conserved shelterin subunit RAP1. In budding yeast, Rap1 binds directly to telomeres and protects them from cNHEJ[14]. However, in mammalian cells, RAP1 is recruited via TRF2 and its role in end protection remains elusive[15–20].

To examine this idea, we used telomere fluorescence in situ hybridization (FISH) to measure chromosome fusions in mouse embryonic fibroblasts (MEFs) deleted for *Apollo* (also known as *Dclre1b*) and/or *Rap1* (also known as *Terf2ip*) (Extended Data Fig. 1a). In agreement with previous studies, *Apollo*-deleted MEFs showed some chromatid fusions that were LIG4-independent and thus were not caused by cNHEJ[13] (Fig. 1a–c). No fusions were observed upon CRISPR-mediated deletion of *Rap1* in Apollo-proficient MEFs, also consistent with previous work[19–21] (Fig. 1a–c). However, when *Rap1* and *Apollo* were deleted together, approximately 15% of telomeres per metaphase were engaged in LIG4-dependent chromosome-type fusions (Fig. 1a,c). A similar effect was induced by mutating the binding sites for Apollo and RAP1 on TRF2, and the resulting fusions were prevented by deleting the inhibitor of DNA damage response (iDDR) motif (Extended Data Fig. 1b–d), which

restores telomeric overhangs in the absence of Apollo[13]. These data strongly suggest that 3′ overhangs and RAP1 can each protect mouse telomeres from cNHEJ. To test whether this is also the case for human telomeres, we repeated the analysis in non-transformed human cells. Unlike in cancer cell lines[22,23] telomere fusions were not observed in *TP53*[−/−] RPE-1 cells lacking *APOLLO* (also known as *DCLRE1B*) (Fig. 1d,e and Extended Data Fig. 1g). However, consistent with the experiments above, 20–30% of telomeres fused in a DNA-PK-dependent manner when RAP1 was also removed (Fig. 1d,e, and Extended Data Fig. 1e–h). A proportion of these fusions involved only one telomere per chromosome end, and in line with the established role of Apollo[10,11], these telomeres had exclusively been replicated as the leading strand (Extended Data Fig. 1g,i). We conclude that in mouse and human cells, telomeres are protected from cNHEJ by two equally effective and parallel pathways: 3′ overhangs and the presence of RAP1.

We focused our attention on the protective function of RAP1. As shelterin co-precipitates with KU and the DNA-PK catalytic subunit (DNA-PKcs) in cell extracts[3,4,24–27], we considered whether RAP1 might prevent cNHEJ by binding directly to DNA-PK. To test this idea, we used DNase I footprinting to examine the position of purified shelterin and DNA-PK on a blunt-ended telomeric template (Fig. 2a and Extended Data Fig. 2a). DNA-PK protected 32 bp from the telomere end, consistent with the assembly of a terminally positioned complex[28] (Fig. 2b). Addition of shelterin reduced the overall efficiency of DNase I cleavage. However, a footprint of precisely 10 bp (the 'shelterin' region) that was not observed with shelterin alone was also visible directly next to DNA-PK. Combined addition of RAP1 and TRF2 (ref. 29), but not

[1]Telomere Biology Laboratory, The Institute of Cancer Research, London, UK. [2]Department of Biomedical and Clinical Sciences, Linköping University, Linköping, Sweden. [3]Structural Biology of Cell Signalling, The Institute of Cancer Research, London, UK. [4]Functional Proteomics, The Institute of Cancer Research, London, UK. [5]Post-translational Modifications and Cell Proliferation, The Institute of Cancer Research, London, UK. ✉e-mail: francisca.lottersberger@liu.se; max.douglas@icr.ac.uk

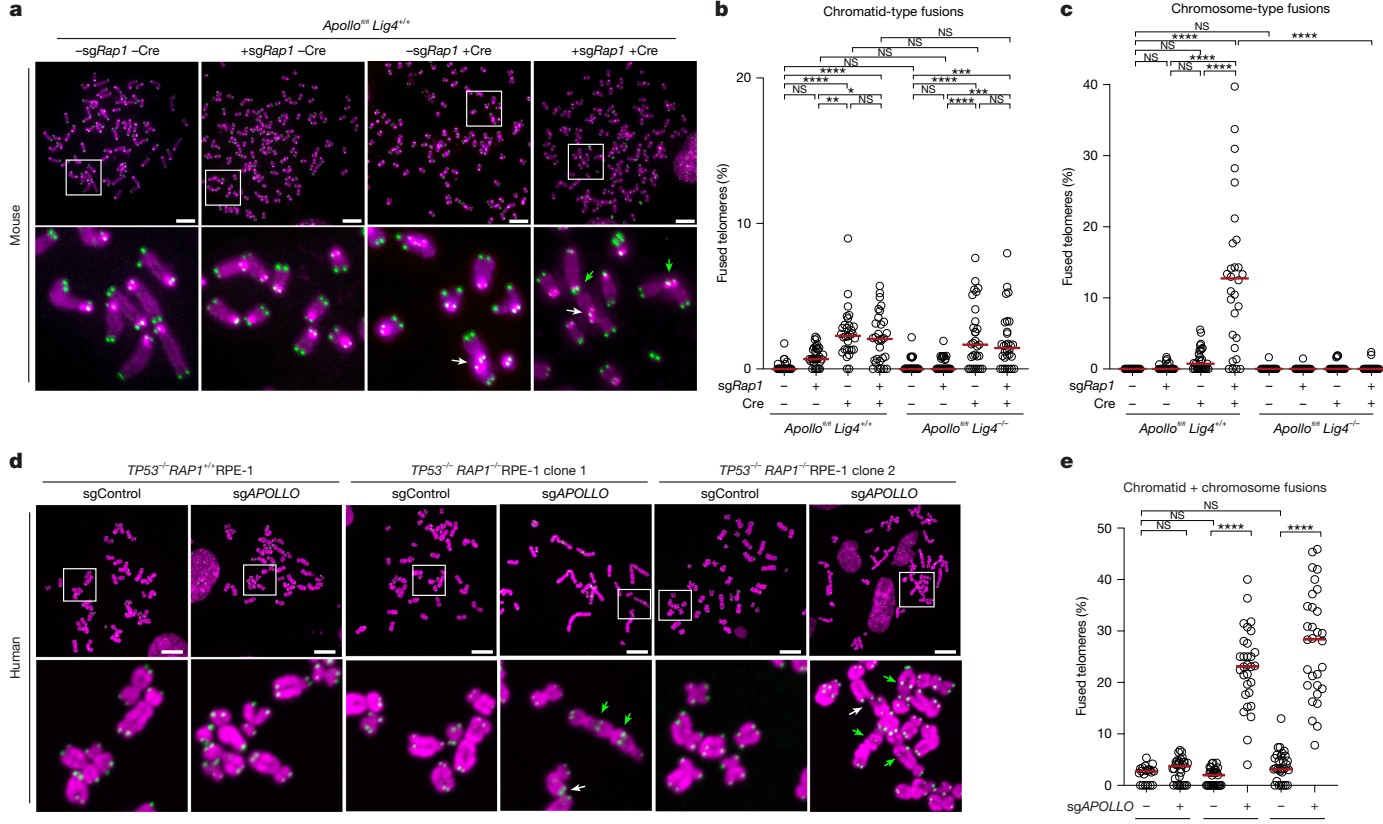

**Fig. 1 | RAP1 and Apollo redundantly prevent cNHEJ at telomeres in mouse and human cells. a**, Representative FISH of metaphase spreads of *Apollo^fl/fl Lig4^+/+* MEFs 108 h after transduction with single guide RNA (sgRNA) targeting *Rap1* (sg*Rap1*) and/or Hit&Run Cre. Telomeres were detected with Cy3-(TTAGGG)₃ (green) and DNA was stained with DAPI (magenta). White and green arrows highlight chromatid-type and chromosome-type fusions, respectively. See also Extended Data Fig. 1a. Scale bars 10 μm. **b,c**, Percentage of telomeres involved in chromatid-type (**b**) or chromosome-type (**c**) fusions per metaphase after removal of Apollo and/or RAP1 as indicated, in the presence or absence of LIG4.

Data from 3 independent experiments, 10 metaphases per experiment (*n* = 30 total), with median. **d**, Representative FISH of metaphase spreads from *TP53^−/− RAP1^+/+* (*RAP1* is also known as *TERF2IP*) or *TP53^−/− RAP1^−/−* RPE-1 cells 120 h after transduction with Cas9 and control sgRNA (sgControl) or sg*APOLLO* as indicated. Scale bars, 10 μm. **e**, Quantification of the percentage of telomeres fused per metaphase after removal of Apollo as described in **d**. Data from 3 independent experiments, 10 metaphases per experiment (*n* = 30 total), with median. See also Extended Data Fig. 1e–g. Ordinary one-way analysis of variance (ANOVA). *P ≤ 0.05, **P ≤ 0.01, ***P ≤ 0.001, ****P ≤ 0.0001; NS, not significant.

each component alone, reproduced this effect (Fig. 2c–e), which was specific to telomeric DNA (Fig. 2f) and was observed over a range of protein concentrations (Extended Data Fig. 2b). Extended Data Fig. 2c shows the same footprint adjacent to DNA-PK was also observed in the presence of a 3′ overhang, but only in the absence of POT1–TPP1, which otherwise prevented DNA-PK assembly.

The data above suggest that DNA-PK bound to telomeric ends can form a sequence-specific complex with TRF2 and RAP1. To examine the architecture of this complex, we determined which domains were required for the DNA-PK-proximal footprint (Fig. 3a and Extended Data Fig. 2d,e). Binding of RAP1 to TRF2 via the RCT (RAP1 C-terminal domain) on RAP1 and the RBM (RAP1-binding motif) on TRF2 (ref. 30), and binding of TRF2 to DNA via the Myb- but not the basic domain was necessary for the extended footprint, suggesting a model in which RAP1 recruitment to DNA by TRF2 is required for the complex to form (Fig. 3b,c and Extended Data Fig. 2f). We tested whether this is the primary function of TRF2 in the assay by fusing RAP1 to the DNA-binding domain of fission yeast Teb1, which recognizes TTAGGG repeats[15]. In the presence of DNA-PK, the Teb1–RAP1 fusion protein generated a 10-bp footprint that was indistinguishable from that of RAP1 in complex with TRF2 and was prevented by cleavage of the Teb1–RAP1 linker (Fig. 3d and Extended Data Fig. 2g,h). Thus, the role of TRF2 in forming the complex is to recruit RAP1 to DNA, and the extended footprint is caused by RAP1, and not by TRF2.

As RAP1 has been proposed to bind KU in cell extracts[25,26], we examined whether purified RAP1 and KU could bind each other in pulldown assays, but did not detect an interaction. However, when RAP1 and KU were mixed in the presence of an amine-specific crosslinker and DNA, we observed a crosslinked species on silver-stained polyacrylamide gels that contained RAP1 and KU as demonstrated by immunoblotting, suggesting that RAP1 mediates a weak interaction with KU (Fig. 3e). Crosslinking efficiency was unaffected by a fragment of TRF2 that contained the RBM and Myb domains, suggesting it is not limited by the proximity of RAP1 to DNA (Extended Data Fig. 2i). Deleting either the BRCT or Myb domain of RAP1 disrupted binding to KU (Fig. 3e) and prevented the extended signal in a footprinting assay (Fig. 3f). To test whether the BRCT domain could be substituted for a different DNA-PK-binding peptide, we fused RAP1(ΔBRCT) to the first BRCT domain of LIG4, which recognizes KU70 (refs. 31,32). Figure 3g shows that adding the LIG4 BRCT domain to RAP1(ΔBRCT) rescued the 10-bp footprint next to DNA-PK, confirming that the necessary function of the RAP1 BRCT domain in forming the complex is recognition of KU.

Although it is possible that one or other of these regions directly protects the 10 bp next to DNA-PK (see below), these data reveal that three molecular interfaces are required for TRF2 and RAP1 to form a complex with DNA-PK: binding of TRF2 to DNA, binding of TRF2 to RAP1, and binding of RAP1 to KU. The requirement for TRF2 to bind DNA explains why the extended footprint is not formed at non-telomeric ends (Fig. 2f), which do not contain a TRF2-binding site.

To determine how this complex might inhibit cNHEJ at telomeres (Fig. 1), we used cryo-electron microscopy (cryo-EM) to examine

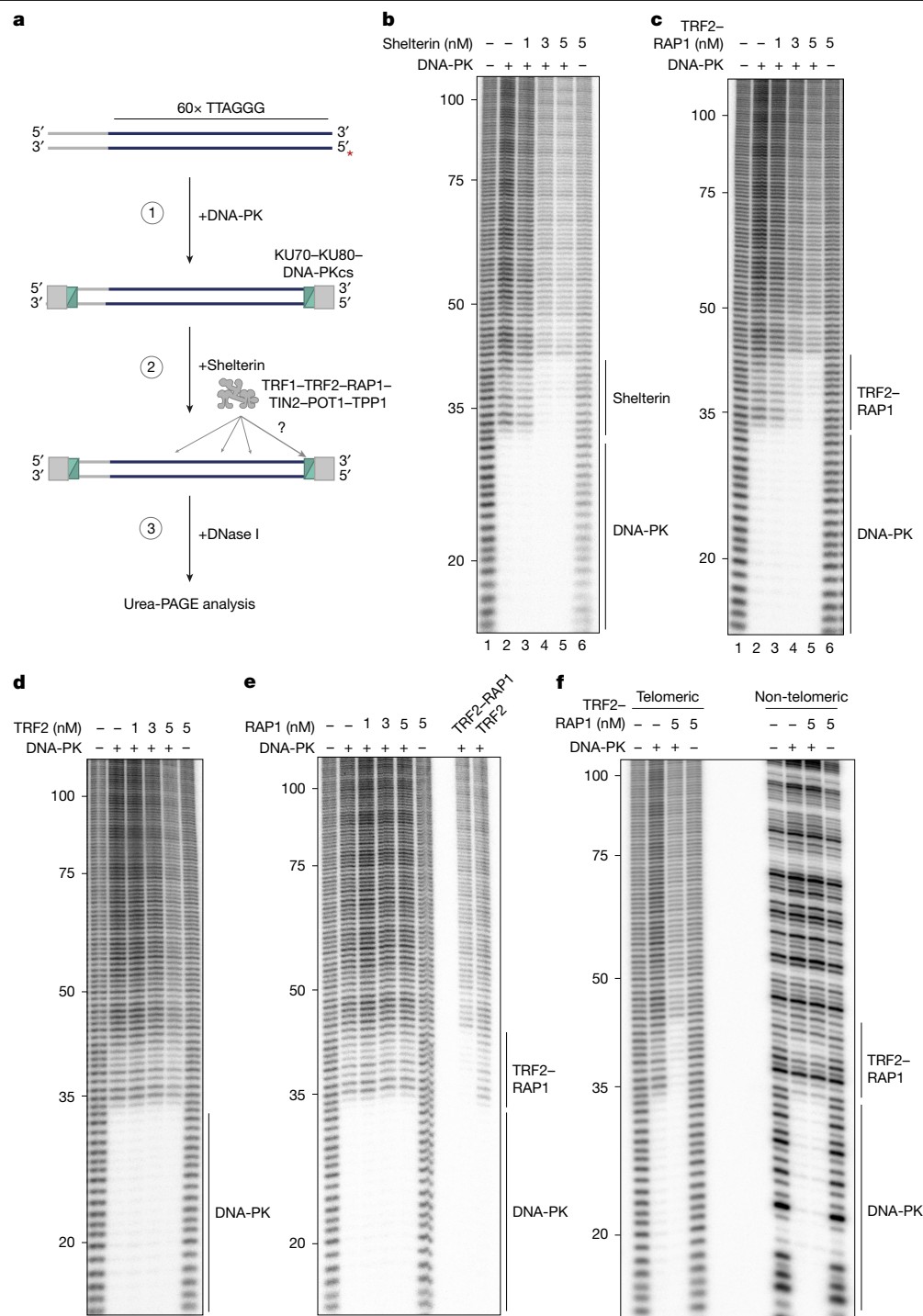

**Fig. 2 | TRF2, RAP1 and DNA-PK form a terminal complex at telomeric DNA ends. a**, Outline of DNase I footprinting experiment. The ³²P-labelled 5′ end is highlighted with a red asterisk. Radiolabelled template is incubated with KU and DNA-PKcs prior to the addition of shelterin, comprising TRF1, TRF2, RAP1, TIN2, POT1 and TPP1. DNase I-digested products are then analysed by denaturing urea polyacrylamide gel electrophoresis (urea-PAGE). **b**–**f**, DNase I footprinting of telomere end-binding complexes formed in the presence of DNA-PK and shelterin (**b**), TRF2 and RAP1 (**c**), TRF2 alone (**d**), RAP1 alone (**e**) TRF2 and RAP1 with telomeric or non-telomeric DNA (**f**). Nucleotides from the 5′ telomeric end indicated. For gel source data see Supplementary Fig. 1.

the structure of TRF2, RAP1 and DNA-PK on telomeric DNA. Consecutive rounds of classification and focused refinement yielded an overall structure at 3.58 Å resolution (Fig. 4a, Extended Data Fig. 3 and Supplementary Video 1). Our map did not contain TRF2, consistent with some inherent flexibility[33] and the notion that its primary role in the complex is to recruit RAP1 to DNA rather than stably bind DNA-PK. The conformation of DNA-PKcs was consistent with previous structures of the unphosphorylated enzyme[34] (Extended Data Fig. 4).

Three additional regions of density were observed beyond the core DNA-PK complex: the first sits on the DNA as it enters KU and was modelled as the KU70 SAP domain (Fig. 4a,b and Extended Data Fig. 5a). The SAP domain is positioned on the minor groove with K575, K595 and K596, which have been proposed to bind DNA[35],

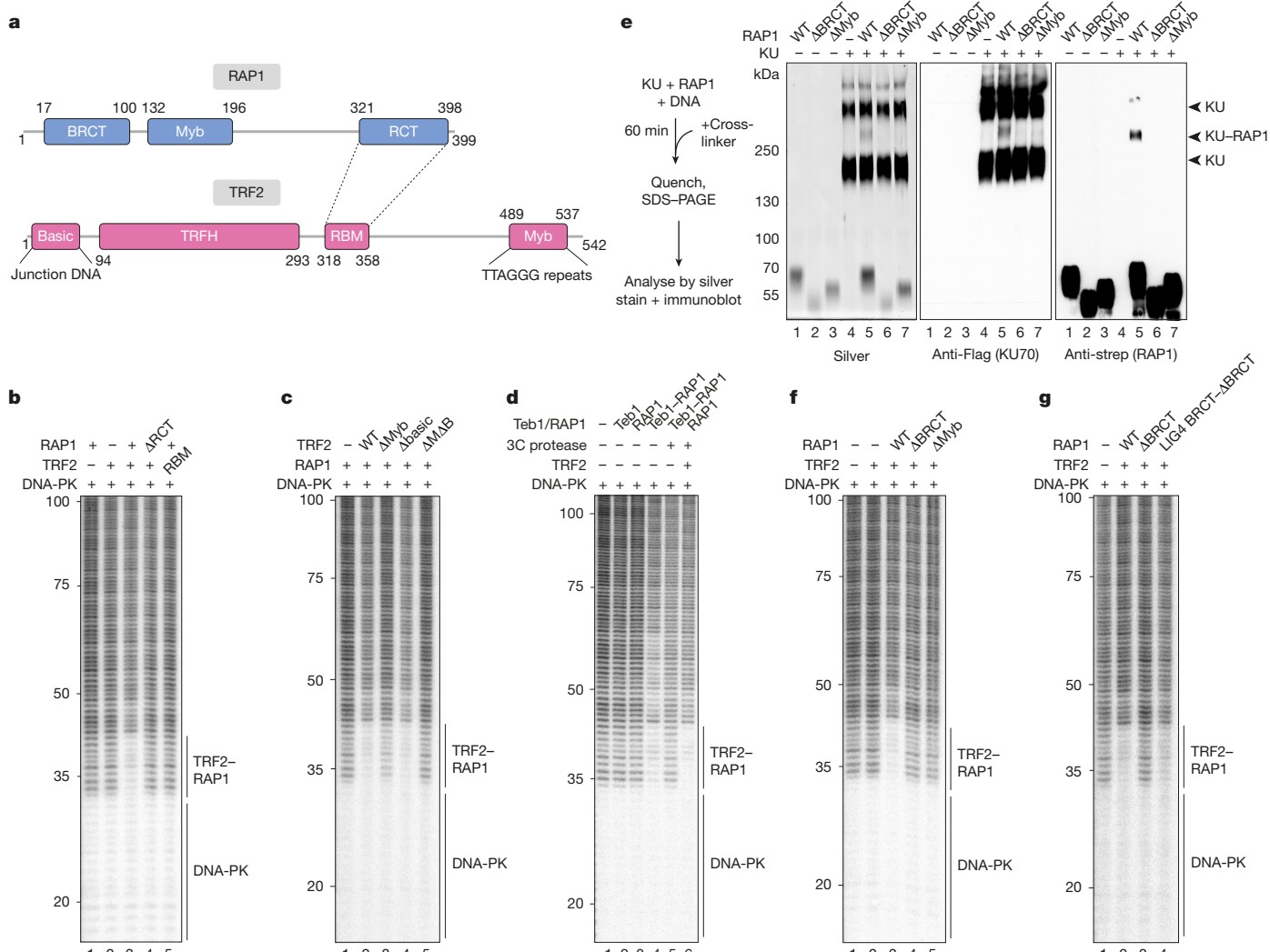

**Fig. 3 | Three distinct interfaces are required for the complex with DNA-PK.**
**a**, Domain organization of RAP1 and TRF2. TRFH, TRF homology domain.
**b–d**, DNase I footprinting of telomere end-binding complexes, testing the
requirement for RAP1 RCT or TRF2 RBM (**b**) TRF2 Myb domain, basic domain or
both Myb and basic domains (ΔMΔB) (**c**), or testing the requirement for TRF2 in
the presence of Teb1, RAP1 or Teb1–RAP1 (**d**). Nucleotides from the 5′ telomeric
end indicated. See Extended Data Fig. 2 for details. WT, wild type. **e**, Protein
crosslinking analysis of RAP1 and KU in the presence of DNA. Proteins were
mixed with crosslinker and reaction products were separated on a denaturing

tris-acetate polyacrylamide gel and analysed by silver staining or immunoblotting
as indicated. Arrowheads mark the position of crosslinked species containing
only KU, or KU and RAP1 as indicated. Bottom and top bands observed with
KU alone are presumed to represent KU dimers and tetramers, respectively.
**f,g**, DNase I footprinting of telomere end-binding complexes, testing the
requirement for BRCT or Myb domains of RAP1 (**f**) and rescue of RAP1(ΔBRCT)
by fusion to the LIG4 BRCT domain (**g**). Nucleotides from the 5′ telomeric end
indicated. For gel source data see Supplementary Fig. 1.

coordinating the phosphate backbone (Extended Data Fig. 5a). The
second region engages the neighbouring 10 bp that are protected
by RAP1 in our footprinting assays and was modelled as the RAP1
Myb domain (Fig. 4a,b and Extended Data Fig. 5b). The Myb domain
adopts a canonical homeodomain arrangement with the recogni-
tion helix inserted into the major groove and a conserved N-terminal
arm[36] formed by R133 reaching into the minor groove bound by the
KU70 SAP (Fig. 4b and Extended Data Fig. 5b–d). Chemical crosslink-
ing mass spectrometry confirms the proximity of the Myb and SAP
regions (Extended Data Fig. 5e,f). This density is surprising because
human RAP1 does not appreciably bind DNA in isolation owing to a
lack of surface-exposed positive charge[29,37,38]. Our structure suggests
that DNA-PK complements this deficit by anchoring the Myb domain
close to DNA through a neighbouring loop bound to the side of KU80
(Extended Data Fig. 5g). The KU70 SAP domain is proposed to bind
other homeodomain proteins[39] and additional contacts with RAP1
may also be in place but are not resolved in our structure. A single

point mutation in the N-terminal arm (R133E) blocked the footprint
next to DNA-PK (Fig. 4e), consistent with the Myb–DNA interaction
protecting this region.

The third region sits on the KU70 vWA domain and was modelled
as the BRCT domain of RAP1 (Fig. 4a,c), positioning a conserved basic
patch on RAP1 next to an acidic patch on KU (Fig. 4d and Extended
Data Fig. 6a). Consistent with our structure, charge reversal muta-
tions in these regions of RAP1 (K39D/R40E/R55E, abbreviated as
RAP1(KR/DE)) or KU (KU70 (D496K/E499R) plus KU80 (D327K),
abbreviated as KU(DE/KR)) prevented the extended signal in a foot-
printing assay (Fig. 4f,g and Extended Data Fig. 6b) and crosslink-
ing of RAP1 to KU (Extended Data Fig. 6c). Remarkably, overlaying
the RAP1 BRCT domain with the LIG4 BRCT domain bound to KU[32]
shows that RAP1 at this position directly occludes the binding site
for LIG4 (Fig. 4h). These data suggest that forming a complex with
DNA-PK, TRF2 and RAP1 may prevent cNHEJ by simply blocking LIG4
recruitment.

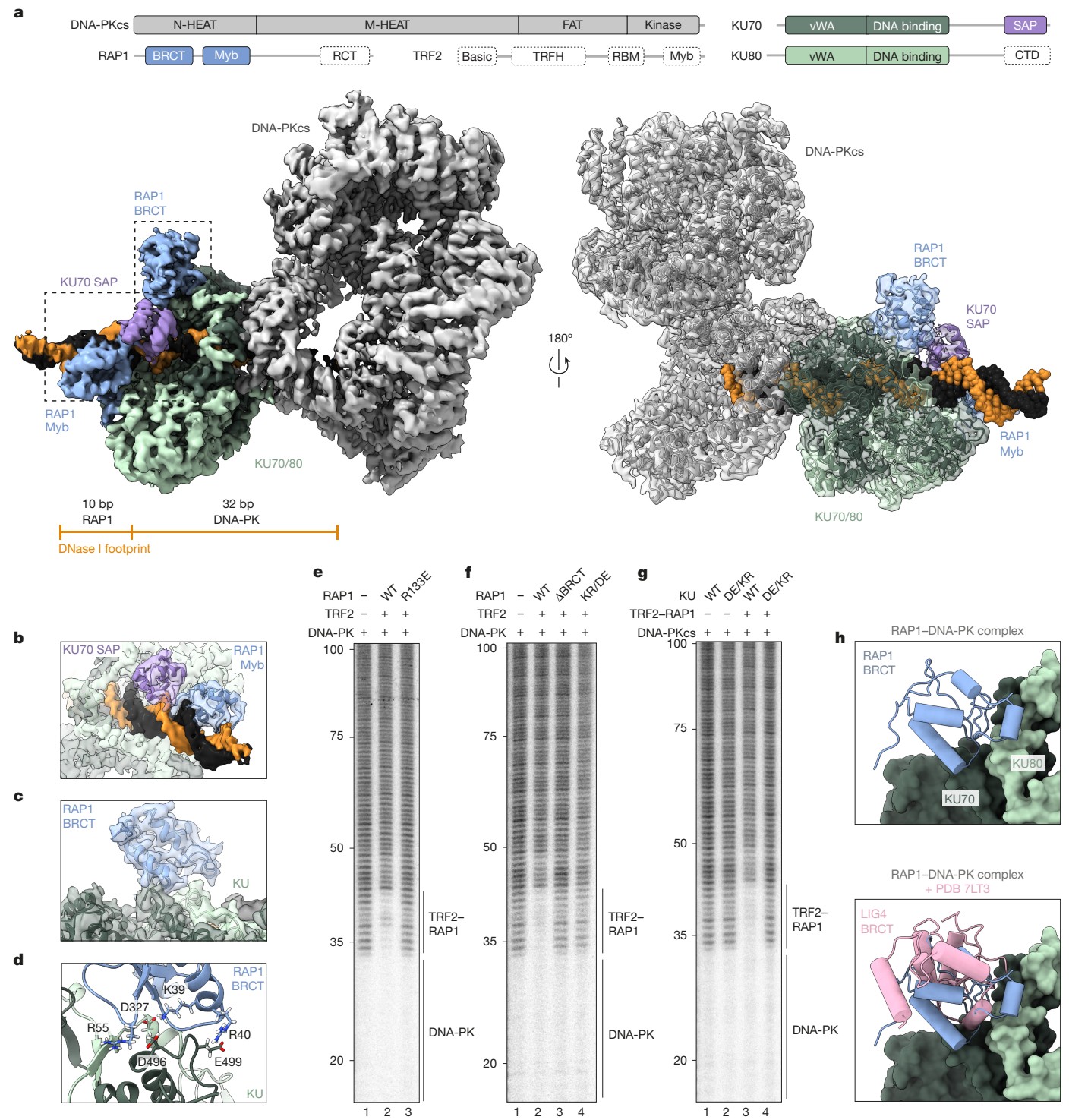

**Fig. 4 | Cryo-EM structure of the RAP1–DNA-PK complex. a**, Bottom, composite electron density map with protein domains coloured as indicated. Top, schematic of proteins used for structure determination. Uncoloured domains were not visualized. CTD, C-terminal domain; FAT, FRAP–ATM–TRRAP domain; M-HEAT, middle Huntington–EF3–PP2A–TOR1 repeat; N-HEAT, N-terminal Huntington–EF3–PP2A–TOR1 repeat; vWA, von Willebrand A domain. **b**, Subsection of the structure in **a**, showing KU70 SAP and RAP1 Myb domains bound to DNA. **c,d**, The BRCT domain of RAP1 bound to the KU70 vWA domain (**c**), highlighting RAP1 and KU residues that mediate the interaction (**d**). **e**–**g**, DNase I footprinting analysis of telomere end-binding complexes with RAP1 variants R133E (**e**) and ΔBRCT and KR/DE (**f**) and KU variant DE/KR (**g**). Nucleotides from the 5′ telomeric end indicated. For gel source data see Supplementary Fig. 1. **h**, Cryo-EM model showing binding of RAP1 BRCT to KU70 and KU80 with (bottom) and without (top) the LIG4 BRCT domain from Protein Data Bank (PDB) structure 7LT3 overlaid.

To test this idea, we developed a pulldown assay in which purified KU was immunoprecipitated after incubation with purified DNA-PKcs, purified XRCC4–LIG4 and a short DNA template (Fig. 5a). DNA-PKcs associated with KU in a DNA-dependent manner, consistent with the assembly of DNA-PK at a DNA end (Extended Data Fig. 7a). XRCC4–LIG4

recruitment was sensitive to mutations in the acidic patch on KU that is required to bind RAP1 and known to bind LIG4 (ref. 32) (Extended Data Fig. 7a). Whereas TRF2 or RAP1 had a negligible effect on the assay individually, recruitment of XRCC4–LIG4 was blocked when they were added together, even when XRCC4–LIG4 was preincubated

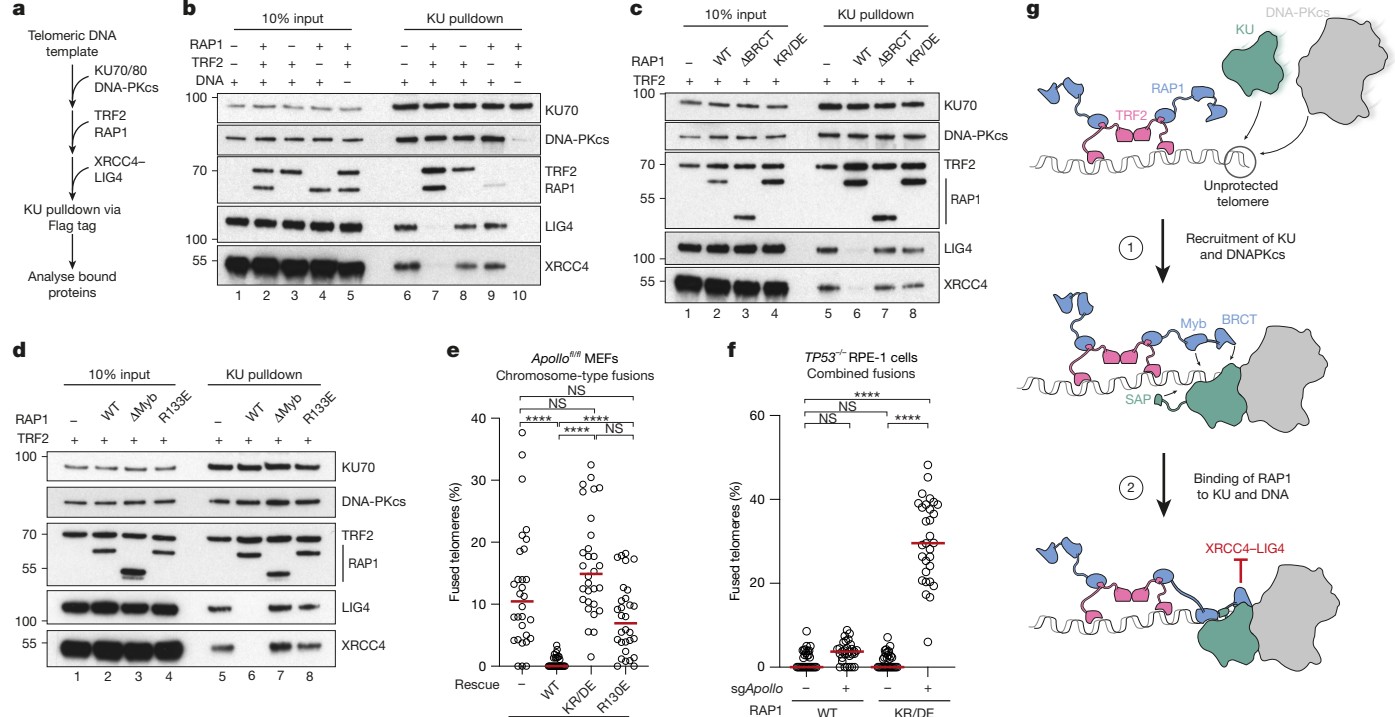

**Fig. 5 | TRF2 and RAP1 prevent cNHEJ by directly blocking recruitment of XRCC4–LIG4 to DNA-PK. a**, Outline of the KU pulldown assay. Details in Methods. **b–d**, KU-bound proteins from reactions containing KU70–KU80 (KU70/80), DNA-PKcs, XRCC4–LIG4, TRF2, RAP1 and template DNA together with wild-type RAP1 (**b**), RAP1(ΔBRCT) or RAP1(KR/DE) (**c**), or RAP1(ΔMyb) or RAP1(R133E) (**d**) were separated by SDS–PAGE and immunoblotted as indicated. TRF2, RAP1 and LIG4 were detected with anti-strep tag antibody, KU70 was detected with anti-Flag antibody. Association of TRF2 with KU is mediated by template DNA. For gel source data see Supplementary Fig. 1. **e**, Percentage of telomeres per metaphase involved in chromosome fusions upon over-expression of mouse RAP1, RAP1(KR/DE) and RAP1(R130E) (equivalent to human RAP1(R133E)) after CRISPR- and Cre-mediated deletion of *Rap1* and *Apollo*, respectively in *Apollo^fl/fl*

*Lig4^+/+* MEFs. Data from 3 independent experiments, 10 metaphases per experiment (*n* = 30 total), with median. Ordinary one-way ANOVA. See Extended Data Fig. 8 for further details. **f**, Percentage of telomeres fused per metaphase upon CRISPR-mediated deletion of *APOLLO* in *TP53^−/−* RPE-1 cells with wild-type RAP1 or RAP1(KR/DE). Data from 3 independent experiments, 10 metaphases per experiment (*n* = 30 total), with median. Statistics as in **e**. See Extended Data Fig. 8 for further details. **g**, Model for the direct inhibition of DNA-PK by TRF2 and RAP1 at mammalian telomeres. When assembled on telomeric DNA, DNA-PK and its associated DNA is bound by the Myb and BRCT domains of RAP1. The BRCT domain acts as a circuit breaker, preventing DNA-PK from engaging LIG4.

with DNA-PK and present in large excess (Fig. 5b and Extended Data Fig. 7b,c). To test whether the ability of TRF2–RAP1 to block LIG4 recruitment required binding to DNA-PK, we repeated the experiment with RAP1(ΔBRCT) or RAP1(KR/DE), which are unable to bind KU (Fig. 4). Inhibition of LIG4 recruitment was not observed under these conditions, but could be restored by fusing the N-terminal BRCT domain of LIG4 to the N-terminus of RAP1(ΔBRCT) (Fig. 5c and Extended Data Fig. 7d). Figure 5d shows that RAP1(R133E) was also defective in preventing recruitment of LIG4, suggesting that multiple contacts between RAP1, KU and DNA are required for this effect.

To examine whether inhibition of LIG4 recruitment is required for RAP1 to prevent cNHEJ at telomeres, *Rap1^−/−Apollo^−/−* MEFs were complemented with the RAP1 mutants examined above. Figure 5e and Extended Data Fig. 8a–c show that RAP1 that is unable to bind DNA-PK and block LIG4 recruitment could not protect telomeres from cNHEJ. Figure 5f and Extended Data Fig. 8d–g demonstrate that this is also the case in human RPE-1 cells.

RAP1 is the most conserved component of eukaryotic chromosome ends[29], yet its role at mammalian telomeres has remained elusive. Our results reveal that it is the defining component of an inhibitory pathway, coordinated by TRF2, in which RAP1 directly and specifically supresses the end-joining function of DNA-PK by preventing the recruitment of LIG4 (Fig. 5g). Binding of RAP1 to DNA may also restrict cNHEJ by preventing KU from translocating inwards, as proposed for budding yeast Rap1 (ref. 40). These findings provide a molecular basis for previous studies that implicate human RAP1 in chromosome end

protection[15–18], offer a mechanism for how cNHEJ can be blocked at telomeres with diverse end structures or lacking t-loops[13,41], and resolve the long-standing paradox that DNA-PK can bind to telomeres and regulate their resection without activating cNHEJ[3,4,23]. By demonstrating that the assembly of DNA-PK can be functionally uncoupled from its end-joining activity through LIG4 recruitment, our study also expands the mechanisms available to regulate pathway choice during double strand break repair.

Apollo and RAP1 are recruited to telomeres via TRF2, which therefore acts as a master regulator of two equally effective pathways to block cNHEJ. Why is RAP1 used when Apollo is apparently sufficient? Processing of telomeres by Apollo depends on DNA-PK[23] and a TRF2-binding motif that is specific to vertebrate homologues[42]. As the RAP1–KU interaction is more broadly conserved across metazoa (Extended Data Fig. 9), we propose that the pathway described here predates the telomeric function of Apollo and enabled DNA-PK at leading strand telomeres to be coopted into a resection role by preventing it from activating cNHEJ. In vertebrate cells, RAP1 will ensure that DNA-PK on blunt leading strand ends cannot engage cNHEJ either before resection occurs or in instances where resection fails. There may also be scenarios in which RAP1 is the primary protective factor. For example, RAP1 deletion alone increases cNHEJ at telomeres in senescent cells[16,43]. Given that only a small number of telomeric repeats are required for TRF2 and RAP1 to prevent LIG4 recruitment in vitro (Fig. 5b), the mechanism described here may be adept at protecting critically short telomeres in these cells from cNHEJ.

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

## Methods

### DNA templates

For DNase I footprinting experiments, a 2.8 kb plasmid containing 360 bp of telomeric DNA was amplified in SURE2 *Escherichia coli* cells grown at 30 °C and extracted using a QIAGEN Plasmid Maxi kit. The plasmid was linearized by BsmFI digestion for 1 h at 37 °C leaving one end with telomeric TTAGGG repeats. The DNA was dephosphorylated with Quick-CIP for 30 min at 37 °C and cleaned up by phenol chloroform extraction and ethanol precipitation. The DNA was subsequently phosphorylated using PNK and [γ-$^{32}$P]ATP for 1 h at 37°C, passed over a G-50 desalting column and phenol chloroform extracted. Labelled DNA was digested with SacI for 1 h at 37 °C to isolate a 390-bp DNA fragment ending in 60 TTAGGG repeats from the rest of the plasmid. Digested DNA was run on a 10% TBE-PAGE gel (Invitrogen) after which the telomeric fragment was excised and gel extracted by shaking in 10 mM Tris pH 8.0, 300 mM NaCl, 1 mM EDTA overnight at 21 °C. The final DNA fragment (PE1; see Supplementary Table 1 for sequence) was precipitated with isopropanol and resuspended in 1× TE buffer. A non-telomeric DNA fragment (PE2; see Supplementary Table 1 for sequence) containing 360 bp of random DNA sequence in place of TTAGGG repeats was prepared through the same procedure using a non-telomeric plasmid. Telomeric DNA templates for experiments in Extended Data Fig. 2c were prepared as above but with initial digestion by Esp3I instead of BsmFI, thereby yielding a different DNA end compatible with ssDNA overhang ligation. To prepare a 15-nt overhang template, dephosphorylated DNA was additionally mixed with oligonucleotide PE5 at a 1:75 DNA:oligonucleotide molar ratio and incubated overnight at 16 °C with T4 DNA ligase (NEB). The ligated template was subsequently cleaned up from excess oligonucleotide using two rounds of HighPrep PCR cleanup beads (MAGBIO), resuspended in TE buffer and γ-$^{32}$P-labelled as above.

Telomeric DNA substrates for cryo-EM, crosslinking and DNA-PK pulldown experiments were prepared by mixing oligonucleotides PE3 and PE4 at a 1:1 molar ratio, heating to 95 °C and cooling to room temperature over 2 h.

### Protein expression

Open reading frames for human KU70/80, RAP1 or TRF2 were codon-optimized for *Spodoptera frugiperda* and cloned into pACEBAC1 vector. The mutants indicated were generated using PCR-based mutagenesis (see Supplementary Table 2 for mutant details). The shelterin genes (*TERF1*, *TERF2*, *RAP1*, *TINF2*, *ACD* (also known as *TPP1*) and *POT1*), codon-optimized for *E. coli*, were synthesized by GenScript and cloned into pACEBAC1, using a nicking cloning system[44]. Vectors were transposed into DH10 MultiBac Competent *E. coli* cells and grown in LB medium overnight at 37 °C shaking at 200 rpm. Bacmid DNA was extracted and used to transfect Sf9 insect cells which were subsequently grown at 27 °C, shaking at 130 rpm over several cell passages. At passage three (P3), 200 ml of High Five insect cells at a 1.5 × 10$^6$ density were infected with baculoviruses for protein expression, incubating at 27 °C with 130 rpm. After 3 days, the cell count was checked, and cells were harvested by centrifugation at 760*g* for 20 min at 4 °C. Cell pellets were then resuspended in PBS, transferred into 50-ml Falcon tubes and pelleted again by centrifugation at 470*g* for 25 min. Supernatants were discarded, the pellets flash frozen in liquid nitrogen and placed at -80 °C until required.

### Nuclear extract preparation

HeLa cell pellets were resuspended in buffer A (10 mM HEPES pH 8.0, 10 mM KCl, 1.5 mM MgCl$_2$, 0.5 mM DTT, 0.5 mM AEBSF) and incubated for 10 min at 4 °C. Following centrifugation at 1,033*g* for 10 min at 4 °C, the pellet was resuspended in two pellet volumes of buffer A and lysed by 20 strokes in a Dounce homogenizer (pestle type B). Lysate was centrifuged at 25,000*g* for 20 min at 4 °C and the pellet (containing nuclei)

was resuspended in 1.3× pellet volumes of buffer C (20 mM HEPES pH 8.0, 420 mM NaCl, 1.5 mM MgCl$_2$, 0.2 mM EDTA, 25% glycerol, 0.5 mM DTT, 0.5 mM AEBSF). After 20 strokes in a Dounce homogenizer (pestle type B), nuclei were incubated for 30 min at 4 °C and centrifuged at 25,000*g* for 30 min at 4 °C. The supernatant was collected and flash frozen in liquid nitrogen.

### Protein purification

**DNA-PKcs.** HeLa cell nuclear extract was diluted into DPKQ buffer (20 mM HEPES pH 7.6, 100 mM NaCl, 2 mM MgCl$_2$, 0.5 mM EDTA, 10% glycerol, 0.5 mM DTT, 0.5 mM AEBSF) and ultracentrifuged at 50,000*g* for 1 h at 4 °C. The supernatant was collected and filtered through a 0.45-μm syringe filter before injection onto a Q-sepharose column equilibrated in DPKQ buffer. The column was washed in DPKQ buffer and proteins eluted over a 100 mM–1 M NaCl gradient. Fractions were spotted onto a nitrocellulose membrane and Western blotted to identify DNA-PKcs-containing fractions, which were pooled, diluted into DPKQ buffer and loaded onto a heparin column equilibrated in the same buffer. The column was washed with DPKQ buffer and proteins eluted over a 100 mM–1 M NaCl gradient. DNA-PKcs-containing fractions were pooled, dialysed into DPKC buffer (50 mM Tris pH 7.3, 0.5 mM EDTA, 5% glycerol, 2.5 mM DTT, 0.5 mM AEBSF) with 50 mM KCl, and loaded onto a dsDNA-conjugated CNBr-activated Sepharose column equilibrated in the same buffer. The column was washed in DPKC buffer with 50 mM KCl and bound proteins were eluted in DPKC buffer with 411 mM KCl. DNA-PKcs-containing fractions were pooled, diluted into DPKC buffer with 100 mM KCl and 0.02% Tween-20 and loaded onto a MonoQ column equilibrated in the same buffer. The column was washed in DPKC buffer with 100 mM KCl and 0.02% Tween-20 and proteins were eluted over a 100 mM–1 M KCl gradient. DNA-PKcs fractions were pooled, diluted into DPKC buffer with 100 mM KCl and 0.02% Tween-20 and loaded onto a MonoS column equilibrated in the same buffer. The column was washed in DPKC buffer with 100 mM KCl and 0.02% Tween-20 and proteins were eluted over a 100 mM–1 M KCl gradient. Final DNA-PKcs fractions were pooled, concentrated using a 100 kDa Amicon Ultra centrifugal filter, flash frozen in liquid nitrogen and stored at −80 °C.

**KU70/80.** Cell pellets were resuspended in 50 mM Tris pH 8.0, 2 mM β-mercaptoethanol with protease inhibitor tablets and incubated stirring for 20 min at 4 °C. Cells were lysed by the addition of 16.7% glycerol and 300 mM NaCl, stirring for 30 min at 4 °C. Lysed cells were ultracentrifuged at 125,000*g* for 30 min at 4°C and the supernatant was incubated with anti-Flag resin for 2 h at 4 °C. Beads were successively washed in Ku buffer (50 mM Tris pH 8.0, 5% glycerol, 2 mM β-mercaptoethanol) with 300 mM NaCl followed by Ku buffer with 150 mM NaCl. Proteins were eluted using the latter buffer supplemented with 0.5 mg ml$^{-1}$ 3× Flag peptide. Proteins were subsequently separated on a Superdex 200 gel filtration column, equilibrated and run using Ku buffer with 150 mM NaCl. Final Ku70/80 fractions were pooled, concentrated using a 30 kDa Amicon Ultra centrifugal filter, flash frozen in liquid nitrogen and stored at −80 °C.

**TRF2 and RAP1.** For TRF2 and RAP1, cell pellets were resuspended in 50 mM HEPES pH 7.6, 1 mM DTT with EDTA-free protease inhibitor tablets (one per 50 ml, Roche) and incubated with stirring for 20 min at 4 °C. Protein was extracted by the addition of 16.7% glycerol and 300 mM NaCl, stirring for 30 min at 4°C. Extract was cleared by ultracentrifugation at 125,000*g* for 30 min at 4 °C. The supernatant was applied to Strep-Tactin XT 4Flow resin equilibrated in shelterin buffer (50 mM HEPES pH 7.6, 500 mM NaCl, 10% glycerol, 1 mM DTT). Beads were washed in shelterin buffer and proteins were eluted with the same buffer supplemented with 10 mM biotin. Proteins were subsequently separated on a Superdex 200 10/300 column equilibrated and run in shelterin buffer. Final TRF2 or RAP1 fractions were pooled,

concentrated using a 30 kDa Amicon Ultra centrifugal filter, flash frozen in liquid nitrogen and stored at −80 °C. To cleave the TEB1–RAP1 fusion protein, 0.24 µM PreScission protease was incubated with 1.2 µM TEB1–RAP1 for 2 h at 4 °C prior to experiments.

**Shelterin.** For the shelterin complex with and without POT1–TPP1, cell pellets containing all 6 shelterin subunits or all subunits except POT1 and TPP1 overexpressed were resuspended in lysis buffer (50 mM Hepes pH 8.0, 300 mM NaCl, 10% glycerol, 1 mM MgCl$_2$, 10 mM beta-mercaptoethanol, 0.1 µl ml$^{-1}$ Base muncher nuclease, 8 µg ml$^{-1}$ Avidin, 1 mM AEBSF and EDTA-free protease inhibitor tablets (one per 50 ml, Roche)) and cells were lysed by sonication. Lysate was cleared by centrifugation at 48,380$g$ for 1 h, 4 °C. Cleared lysate was passed through a 0.45-µm filter and applied to a StrepTrap column equilibrated with StrepTrap wash buffer (50 mM Hepes 8.0, 300 mM NaCl, 10 glycerol and 1 mM tris-2-carboxyethyl phosphine (TCEP)), which was washed with 20 column volumes StrepTrap wash buffer prior to elution with 5 column volumes StrepTrap wash buffer supplemented with 10 mM D-desthiobiotin. Eluate (the 0.5 ml eluate containing the highest concentration of shelterin) was applied to a Superose 6 10/300 column preequilibrated in 50 mM Hepes pH 8.0, 300 mM NaCl, 10% glycerol and 1 mM TCEP, and run in the same buffer. The desired fractions were aliquoted and flash frozen for storage. Further characterization of the full shelterin complex will be reported elsewhere.

**XRCC4–LIG4.** Cell pellets were resuspended in 50 mM HEPES pH 7.6, 1 mM DTT, 1 mM EDTA with EDTA-free protease inhibitor tablets (Roche, 1 per 50 ml buffer) and incubated with stirring for 20 min at 4 °C. Proteins were extracted by the addition of glycerol to 16.7% and NaCl to 300 mM with stirring for 30 min at 4 °C. Extract was centrifuged at 41,656$g$ for 30 min at 4 °C and the supernatant was applied to Strep-Tactin XT 4Flow resin equilibrated in X4 buffer (50 mM HEPES pH 7.6, 300 mM NaCl, 10% glycerol, 1 mM DTT, 1 mM EDTA). Beads were washed in X4 buffer and proteins were eluted with the same buffer supplemented with 30 mM biotin. Protein fractions were pooled, the NaCl concentration diluted to 100 mM then applied to a 1 ml heparin column and subjected to a linear gradient from 0.1 M to 1 M NaCl over 20 column volumes. Proteins were subsequently separated on a Superdex 200 gel filtration column equilibrated and run in 100 mM NaCl buffer (50 mM HEPES pH 7.6, 100 mM NaCl, 10% glycerol, 1 mM DTT, 1 mM EDTA). Final XRCC4–LIG4 fractions were pooled, concentrated using a 30 kDa Amicon Ultra centrifugal filter, flash frozen in liquid nitrogen and stored at −80 °C.

### Nano differential scanning fluorimetry
Purified proteins as indicated were analysed by nano differential scanning fluorimetry to assess thermal protein stability using a Tycho NT.6 (Nanotemper) with 10 µl capillaries, monitoring fluorescence at 330 and 350 nm over a 35–95 °C temperature ramp (30 °C min$^{-1}$).

### DNase I footprinting
[γ-$^{32}$P]-labelled PE1 (telomeric) or PE2 (non-telomeric) template (2 nM) was mixed with 30 nM DNA-PKcs and 50 nM KU70/80 in 25 mM HEPES pH 7.6, 80 mM KCl, 1.5 mM CaCl$_2$, 1.5 mM MgCl$_2$, 5% glycerol, 50 µg ml$^{-1}$ BSA, 2 mM DTT and incubated on ice for 5 min. 5 nM shelterin or TRF2–RAP1 (dimer concentration of TRF2) was added to DNA-bound DNA-PK and incubated at 37 °C for 10 min. Nuclease cleavage was initiated by addition of DNase I to 0.5 U ml$^{-1}$ and the reactions were incubated for a further 2 min at 37 °C before quenching with 25 mM EDTA, 0.2% SDS, 0.2 mg ml$^{-1}$ Proteinase K. Following incubation at 37 °C for 10 min, samples were extracted with phenol chloroform, ethanol precipitated and resuspended in 2 µl 99% formamide, 5 mM EDTA, bromophenol blue. Samples were boiled for 2 min and run on an 8% urea-PAGE sequencing gel in 1× TBE. Gels were subsequently dried and exposed to a BAS-MS imaging plate before phosphor imaging using a Typhoon Biomolecular Imager (Amersham). Images were analysed in ImageJ2 (v.2.14.0) and

Adobe Photoshop (v.25.0.0) and figures were prepared using Adobe Illustrator (v.25.0.0). Footprinting experiments in Extended Data Fig. 2b,c were prepared as above but with higher protein and/or DNA concentrations as indicated in figure legend. Extended Data Fig. 2c was also performed with coincident addition of shelterin, KU and DNA-PKcs.

### Electrophoretic mobility shift assay
[γ-$^{32}$P]-labelled PE1 (telomeric) template (1 nM) was mixed with 2, 15 or 40 nM TRF2 in 25 mM HEPES pH 7.6, 80 mM KCl, 1.5 mM CaCl$_2$, 1.5 mM MgCl$_2$, 5% glycerol, 50 µg ml$^{-1}$ BSA, 2 mM DTT. 10 µl reactions were incubated at 37 °C for 15 min. Samples were supplemented with 1% sucrose, Orange G and run on a 1.5% agarose gel in 0.5× TBE. Gels were dried and analysed by phosphor imaging as with DNase I footprinting experiments.

### Crosslinking
KU70/80 (200 nM) was mixed with 200 nM RAP1 in 20 mM HEPES pH 7.6, 200 nM NaCl, 2 mM MgCl$_2$, 0.5 mM EDTA, 10% glycerol, 0.5 mM DTT, 0.5 mM AEBSF and incubated for 5 min at 4 °C. Samples were supplemented by 100 nM annealed PE3/PE4 DNA substrate and incubated for 10 min at 4 °C. Proteins were crosslinked by addition of 2 mM DSSO and incubated for 60 min at room temperature. Reactions were stopped with 20 mM Tris pH 7.6. Samples were run on a Criterion XT 3-8% Tris-Acetate PAGE gel in 1× XT Tricine and analysed by silver staining (SilverQuest, Invitrogen) or immunoblotting, probing for KU70 (Flag) or RAP1 (strep) (see 'Antibodies' and 'Immunoblotting'). The crosslinking experiment in Extended Data Fig. 2i was performed as above with 200 nM TRF2(ΔBΔTRFH) (with cleaved off strep tag) added together with RAP1.

### DNA-PK pulldown
Annealed PE3/PE4 DNA substrate (5 nM) preincubated with a twofold excess of streptavidin (IBA) was incubated with 10 nM KU70/80 and 15 nM DNA-PKcs for 3 min at 30 °C in 20 mM HEPES pH 7.6, 80 mM KCl, 5% glycerol, 0.01% NP-40 and 1 mM DTT in a final volume of 25 µl in protein low bind tubes (Alpha Labs). 5 nM TRF2–RAP1 complex (dimer concentration of TRF2) was added, and after a further 3 min, 30 nM XRCC4–LIG4 complex was added. After a further 3 min, the complete reaction was added to 1 µl equivalent of anti-Flag M2 magnetic beads (Sigma), and the mixture incubated at 4 °C with shaking for 30 min. Beads were pelleted on a magnetic rack, washed 3× with 50 µl 25 mM HEPES pH 7.6, 80 mM KCl, 10% glycerol, 0.02% NP-40 and 1 mM DTT with a brief vortex included for each wash. Beads were resuspended in wash buffer supplemented with 0.25 mg ml$^{-1}$ 3× Flag peptide and incubated for 20 min at 18 °C with shaking. Eluted proteins were supplemented with SDS loading buffer, run on a 4–12% TGX precast gel (Bio-Rad), transferred onto nitrocellulose membrane at 80 V for 90 min and detected by immunoblotting with the antibodies indicated (see 'Antibodies' and 'Immunoblotting'). Pulldown experiments in Extended Data Fig. 7b,c were prepared as above but with 30–120 nM XRCC4–LIG4 as indicated.

### Cryo-EM sample preparation
Annealed PE3/PE4 DNA substrate (see 'DNA templates') was mixed 1:1 with streptavidin in TE buffer and incubated for 30 min at room temperature. Streptavidin-bound DNA was diluted to 250 nM in 40 mM HEPES pH 7.6, 100 mM NaCl, 3 mM MgCl2, 1 mM DTT, 2.5% glycerol and incubated with 250 nM KU70/80 and 250 nM DNA-PKcs for 10 min at 4 °C. 250 nM TRF2–RAP1 (dimer concentration of TRF2) was added and incubated for another 5 min at 4 °C. Samples were supplemented with 0.05% CHAPS prior to cryo-EM grid preparation.

### Cryo-EM data acquisition and image processing
Copper R1.2/1.3 grids (300-mesh, Quantifoil) were coated with a thin layer or carbon and glow-discharged at 15 mA for 30 s (PELCO easiGlow).

Three microlitres of sample was applied to glow-discharged grids and incubated for 5 s followed by blotting for 3 s using a Vitrobot Mark IV (Thermo Scientific) operated at 4 °C and 100% humidity. Grids were subsequently plunge-frozen in liquid ethane. Cryo-EM data were acquired at 200 kV on a Glacios Cryo-TEM (Thermo Scientific) equipped with a Falcon 4i Direct Electron Detector (Thermo Scientific). In total, 30,604 movies with 30 frames were collected at 150,000× magnification (0.94 Å pixel size) with a total electron dose of 50 e$^-$ Å$^{-2}$ and a defocus range of −1.0 to −2.5 μm (see Extended Data Table 1). Subsequent image processing was performed in cryoSPARC (v4.3.1)[45]. Movies were motion corrected using patch alignment with all frames followed by patch contrast transfer function estimation. Particles were picked through automated template-based picking (template EMD-6803) and extracted with 4× binning using a box size of 96 pixels. Following 2D classification 1,464,362 DNA-PK particles were selected to reconstruct an ab initio 3D model, subsequently used as a starting model for heterogeneous refinement using 5 classes. The most prominent DNA-PK class was selected and subjected to homogeneous refinement followed by local refinement using a focus mask encompassing the KU70/80–DNA core. A 3D classification without alignment using the same mask and ten classes was performed to identify particles containing additional RAP1 and KU densities. Classes lacking either the RAP1 BRCT the RAP1 Myb domain or the KU70 SAP domain were excluded. A total of 526,885 selected particles was re-extracted unbinned with a 384-pixel box size and a 3D map was reconstructed through homogeneous refinement. Following global contrast transfer function refinement a structure of the full DNA end-binding complex was resolved to 3.58 Å using homogeneous refinement. The KU–RAP1–DNA core was locally refined to 3.32 Å using a focus mask excluding DNA-PKcs. The DNA-PKcs–DNA density was similarly refined using a mask excluding the KU–RAP1–DNA core and subjected to 3D classification without alignment using the same mask. Some flexibility in DNA-PKcs conformation was observed and classes of the most prominent conformation containing 370,172 particles were selected. A DNA-PKcs–DNA structure from these particles was resolved to 3.40 Å by homogeneous refinement followed by local refinement using a DNA-PKcs focus mask. Maps for the full end-binding complex and the locally refined densities were sharpened in cryoSPARC and combined using Phenix (v1.20.1) combine_focused_maps[46]. The composite map (EMD-19065) was subsequently used for model building in Coot[47] and figure generation in ChimeraX[48].

## Model building and validation
Molecular models for human DNA-PK (PBD 7K1K), RAP1 Myb (PDB 1FEX) and KU70 SAP (PDB 1JJR) were docked into the cryo-EM map using the Fit in Map command in ChimeraX[48]. The RAP1 BRCT domain from a KU–RAP1 AlphaFold model (see 'AlphaFold modelling') was similarly docked into the cryo-EM map. Models were refined against the map using Namdinator[49] followed by manual inspection in Coot[47]. Unoccupied protein densities and nucleic acids were modelled de novo. The resulting model was iteratively refined using Phenix (v1.20.1) real_space_refinement[50] with geometry and secondary structure restraints followed by manual adjustment in Coot. The quality of the final atomic model (PDB 8RD4) was evaluated by MolProbity[51] in Phenix (see Extended Data Table 1).

## AlphaFold modelling
Full-length human RAP1, KU70 and KU80 were analysed using Alpha-Fold 3 on the online AlphaFold server, with the top ranked prediction shown. For Extended Data Fig. 9, the additional sequences analysed were as follows. *Salmo salar*: NP_001133439.1, XP_014030197.1 and XP_045561280.1; *Strongylocentrotus purpuratus*: XP_030845408.1, XP_030843748.1, XP_001198957.2; *Nematostella vectensis*: XP_001641354.1, EDO36674.1, EDO44451.1; *Trichoplax adhaerens*: XP_002117640.1, XP_002117043.1, XP_002112721.1.

## Protein alignments
Pre-computed protein alignments were analysed using the ProViz online tool[52].

## Antibodies
Human DNA-PKcs was detected with antibody 18-2 (Invitrogen MA5-13238) at 1:100 dilution, human RAP1 with antibody A300-306A (Bethyl Laboratories) at 1:4,000 dilution, human TRF2 with antibody D1Y5D (Cell Signaling 13136) at 1:1,000 dilution, human Apollo with antibody HPA064934 (Atlas Antibodies) at 1:100 dilution and human α-tubulin with antibody T9026 (Sigma) at 1:1,000 dilution. Recombinant human KU70 was detected via an N-terminal Flag tag with antibody M2 (Sigma F1804) at 1:1,000 dilution. Recombinant human LIG4, TRF2 and RAP1 were detected via a dual strep tag using antibody ab76949 (Abcam) at 1:1,000 dilution. Recombinant human XRCC4 was detected with antibody C-4 (Santa Cruz sc-271087) at 1:500 dilution. Primary antibodies for human proteins were detected with goat anti-rabbit IgG–horseradish peroxidase (HRP) (Cell Signalling 7074) or horse anti-mouse IgG–HRP (Cell Signalling 7076) secondary antibody. Mouse RAP1 was detected with antibody D9H4 (Cell Signalling 5433) at 1:1,000 dilution, mouse TRF2 with antibody D1Y5D (Cell Signaling 13136) at 1:1,000 dilution and mouse β-actin with antibody 8H10D10 (Cell Signaling 3700) at 1:4,000 dilution followed by donkey anti-rabbit IgG–HRP (NA934V, Cytiva), goat anti-rabbit IgG–HRP (31460, Invitrogen) or goat anti-mouse IgG–HRP peroxidase (31430, Invitroen) secondary antibody.

## Cell lines and viral gene delivery
SV40-LT *Apollo$^{fl/fl}$ Lig4$^{+/+}$, Apollo$^{fl/fl}$ Lig4$^{-/-}$* and *Trf2$^{fl/fl}$Rosa26$^{cre-ERT1}$* MEFs have been previously described[10,53]. hTERT immortalized RPE-1 cells have been previously described[54]. All MEFs were immortalized with pBabeSV40LargeT and cultured in Dulbecco's Modified Eagle Medium (DMEM, Cytiva) supplemented with 15% fetal bovine serum (FBS, Gibco), non-essential amino acids (Cytiva), L-glutamine (Cytiva), penicillin-streptomycin (Cytiva), 50 μM β-mercaptoethanol (Sigma). 293 T and Phoenix eco cells (ATCC) were cultured in DMEM (Cytiva) supplemented with 10% HyClone Bovine Calf Serum (Cytiva), non-essential amino acids (Cytiva), L-glutamine (Cytiva), and penicillin-streptomycin (Cytiva). RPE-1 cells were cultured in DMEM/F12 medium supplemented with 10% (v/v) FBS, 1% (v/v) penicillin-streptomycin, 1% Glutamax, 0.5 μg ml$^{-1}$ Amphotericin B and 0.26% sodium bicarbonate. To generate *TP53$^{-/-}$* RPE-1 clones by CRISPR–Cas9 mediated gene editing, cells were electroporated with Cas9–sgRNA ribonucleoparticles targeting the sequences 5′-AAATTTGCGTGTGGAGTATT-3′ and 5′-TCCACTCGGATAAGATGCTG-3′[55] using the Neon Transfection system as described[56]. After 4 days, single cells were sorted into 96-well plates containing medium supplemented with 10 μM nutlin-3a. After 14 days, surviving clones were expanded, and p53 status was analysed by immunoblotting and sequencing of the *TP53* locus as described[55]. CRISPR–Cas9 mediated editing of human *RAP1* in *TP53$^{-/-}$* RPE-1 cells was carried out using phosphorothioated single-stranded DNA repair templates (ssODN) and selection for positive integrands by ouabain as described[57,58]. In brief, *RAP1* guide RNA 5′-GGCCCAGCCCGGCCAAGCGT-3′ was cloned into the BspI site of Addgene vector 86613. Repair templates for *ATPA1* and *RAP1* were synthesized by Integrated DNA Technologies (IDT) with the following sequences: *ATP1A1*: C*A*ATGTTACTGTGGATTGGAGCGATTCTTTGTTT CTTGGCTTATAGCATCAGAGCTGCTACAGAAGAGGAACCTCAAAACGA TGACGTGAGTTCTGTAATTCAGCATATCGATTTGTAGTACACATCAGATAT C*T*T; *RAP1*: C*A*TTCCTCGACTCTGTTCGTGAGGGACGACGGCAGCTCC ATGTCCTTCTACGTGCGGCCCAGCCCGGCCGACGAGCGCCTCTCGACG CTCATCCTGCACGGCGGCGGCACGGTGTGCGAGGTGCAGGAGCCCG GGGCCGTGCTGCTGGCCCAGCCCGGGGAGGCGCTGGCCGAGGCCTC GGGTGATTTCATCTCCACG*C*A, where * denotes a phosphorthiolated base. To generate RAP1(KR/DE) clones, 300,000 *TP53$^{-/-}$* RPE-1 cells

were electroporated with the Neon Transfection System using a 10 μl tip and two pulses at 1,350 V and 20 ms with 500 ng *RAP1* guide RNA/Addgene plasmid #86613, 2 pmol of *ATP1A1* ssODN and 6 pmol *RAP1* ssODN. To generate *RAP1*[−/−] clones, the procedure was repeated omitting *RAP1* ssODN. After 3–4 days, cells were expanded into 15-cm dishes and treated with 0.25 μM ouabain, followed by isolation of single clones 7–12 days after drug selection. Genomic DNA was prepared using EZNA Tissue DNA kit according to the manufacturer's instructions. *RAP1* was amplified by PCR using oligonucleotide sequences AGTGCTG CGCTTCGCGGC and CGCCTTCCGCTTGAGCTTCTG. Editing was analysed by restriction digestion with SalI and Sanger sequencing, and positive clones were single cell sorted and expanded prior to freezing. For retro or lentiviral transduction, a total of 20 μg of plasmid DNA was transfected into Phoenix eco or 293 T cells, respectively, using CaPO4 precipitation. The viral supernatant was filtered through a 0.45-μm filter, supplemented with 4 μg ml[−1] polybrene, and used for the transduction of target cells. Lentiviral particles containing the sgRNA against mouse *Rap1* (target: 5′-GCAGTCTAGGATGTACTGCG-3′) in lentiCRISPR v2 (Addgene plasmid #52961, a gift from F. Zhang) were introduced into target MEF cells with three infections per day (6–12 h intervals) over 2 days, followed by 2 days in 2–4 μM Puromycin or 340 μM Hygromycin. The same approach was used to target human *Apollo* in *TP53*[−/−] RPE-1 cells with lentiviral particles containing lentiCRISPR v2 with the guide sequence 5′-CTGGTTCCAACGCAGCATGT-3′[23], or non-targeting control sequence 5′-CGCCAAACGTGCCCTGACGG-3′. For MEF experiments, Cre was induced with three infections per day (6–12 h intervals) over two days with pMMP Hit&Run Cre retrovirus produced in Phoenix eco cells. Time-point 0 was set 12 h after the first Hit&Run Cre infection. For the *Rap1*-complementation assay, *Apollo*[fl/fl] MEFs were transduced with retroviral particle containing *Rap1*, *Rap1*[KR/DE] or *Rap1*[R130E] cloned in Plpc vector for a total of 4 infections at 6–12-h intervals, selected for 2–3 days in 2–4 μM Puromycin, transduced with the sgRNA against mouse Rap1 cloned in LcV2-Hygro (Addgene plasmid #91977, a gift from J. Mendell), selected for 2 days in 340 μM Hygromycin and then transduced with pMMP Hit&Run Cre as previously described. For the *Trf2*-complementation assays, Trf2[fl/fl]*Rosa26*[cre-ERT1] MEFs were transduced with retroviral particle containing the *Trf2* mutants, selected for 2 days in 2–4 μM Puromycin and then treated with 1 μM 4-OHT for 24 h. Time-point 0 was set at the time of 4-OHT addition. All cell lines in this study routinely tested negative for mycoplasma contamination. RPE-1 cells were validated by whole-genome sequencing, while MEFs were automatically genotyped after isolation by TransnetYX for the presence of *Apollo* or *Trf2 flox* alleles and/or ligase 4 deletion or RsCre.

### Generation of *Trf2*-mutant alleles

PCR was used to delete the RAP1-binding motif (RBM) or insert the S367A mutation into MYC-tagged *Trf2*, *Trf2*[F120A], *Trf2*[ΔiDDR] and *Trf2*[F120AΔiDDR] alleles cloned in pLPC retroviral vector[13] using the following primers: *Trf2*[ΔRBM]-F:5′-AATCTGGCATCCCCATCATCAC-3′; *Trf2*[ΔRBM]-R:5′-TCTGCTTG GAGGCTCTCTAAG-3′; *Trf2*[S367A]-F: 5′-GCGCCAGCCCACAAACACAA GAGACC-3′; *Trf2*[S367A]-R: 5′-TGATGGGGATGCCAGATTAGCAAG-3′.

### Fluorescence in situ hybridization

Telomere FISH on mouse and human cells was performed as previously described[59]. In brief, cells were treated with 0.2 μg ml[−1] Colcemid (Biowest/Roche) for 2 h before collection by trypsinization. Collected cells were swollen in a hypotonic solution of 75 mM KCl at 37 °C for 10–20 min before fixation in methanol:acetic acid (3:1) overnight at 4 °C. Cells were dropped onto glass slides and allowed to age overnight. The slides were then dehydrated through an ethanol series of 70%, 95% and 100% and allowed to air dry. Telomere ends were hybridized with Cy3-OO-(TTAGGG)₃ in hybridization solution (70% formamide, 1 mg ml[−1] blocking reagent (1109617601, Roche), and 10 mM Tris-HCl pH 7.2) for 2 h following an initial 5–10 min denaturation step at 80 °C, washed twice with 70% formamide; 0.1% BSA; 10 mM Tris-HCl, pH 7.2 for

15 min each, and thrice in 0.08% Tween-20; 0.15 M NaCl; 0.1 M Tris-HCl, pH 7.2 or PBS for 5 min each. For MEFs, chromosomal DNA was counterstained with the addition of DAPI (D1306, Invitrogen) to the second wash. Slides were left to air dry and mounted in antifade reagent (Prolong Gold Antifade P36934, Fisher). For RPE-1 cells, air-dried slides were mounted in DAPI-supplemented Vectashield mounting medium (Vector laboratories) Micrographs for mouse cell experiments were collected on a DeltaVision RT microscope, Micrographs for RPE-1 cell experiments were collected on a Zeiss Axio Observer Z1 Marianas TM microscope (operated with 3i SlideBook) equipped with a CSU-X1 spinning disk (Yokogawa). Metaphases were analysed in FIJI and scored fusions were plotted using GraphPad Prism. Figures were prepared in Adobe Illustrator (v25.0.0). CO-FISH analysis of RPE-1 cells was performed as previously described[23] with the following modifications: RPE-1 cells were treated with BrdU:BrdC for 14 h, slides were treated with UV for 10 min using a Blak ray model UV-21 365 nm handheld lamp at a distance of 8 cm and telomeres were detected with Cy3-OO-(TTAGGG)₃ and FITC-OO-(CCCTAA)₃.

### Immunoblotting

Cells were lysed in 2× Laemmli buffer at $5 \times 10^3$ or $1 \times 10^4$ cells per μl and the lysate was denatured for 10 min at 95 °C before shearing with an insulin needle or sonication. Lysate equivalent to $1–2 \times 10^5$ cells was resolved using SDS–PAGE and transferred to a nitrocellulose membrane. For mouse cell experiments, Western blot was performed with 5% milk in PBS containing 0.1% (v/v) Tween-20 (PBS-T). For RPE-1 cell experiments, Western blotting was performed in TBS buffer supplemented with 0.1% Tween-20. Additional reagents are described in the 'antibodies' section. Immunoblots were developed using chemiluminescence western blotting detection reagents (Cytiva or Cell Signalling or Millipore) and imaged on a ChemiDoc (Bio-Rad) imaging system or using Amersham Hyperfilm MP (Cytiva) and a CURIX 60 processor (AGFA). Images were analysed in Adobe Photoshop (v25.0.0) and figures were prepared using Adobe Illustrator (v25.0.0).

### Chemical crosslinking mass spectrometry analysis

Complex assembly and crosslinking with DSSO was performed as described in 'Crosslinking'. After the crosslinking reaction, triethylammonium bicarbonate buffer (TEAB) was added to the sample at a final concentration of 100 mM. Proteins were reduced and alkylated with 5 mM TCEP and 10 mM iodoacetamide simultaneously and digested overnight with trypsin at final concentration 50 ng μl[−1] (Pierce). Sample was dried and peptides were fractionated with high-pH Reversed-Phase (RP) chromatography using the XBridge C18 column (1.0 × 100 mm, 3.5 μm, Waters) on an UltiMate 3000 HPLC system. Mobile phase A was 0.1% v/v ammonium hydroxide and mobile phase B was acetonitrile, 0.1% v/v ammonium hydroxide. The peptides were fractionated at 70 μl min[−1] with the following gradient: 5 min at 5% B, up to 15% B in 3 min, for 32 min gradient to 40% B, gradient to 90% B in 5 min, isocratic for 5 min and re-equilibration to 5% B. Fractions were collected every 100 s, SpeedVac dried and pooled into 12 samples for mass spectrometry analysis. Liquid chromatography–mass spectrometry analysis was performed on an UltiMate 3000 UHPLC system coupled with the Orbitrap Ascend Mass Spectrometer (Thermo). Each peptide fraction was reconstituted in 30 μl 0.1% formic acid and 15 μl were loaded to the Acclaim PepMap 100, 100 μm × 2 cm C18, 5 μm trapping column at 10 μl min[−1] flow rate of 0.1% formic acid loading buffer. Peptides were then subjected to a gradient elution on the Acclaim PepMap (75 μm × 50 cm, 2 μm, 100 Å) C18 capillary column connected to the EASY-Spray source at 45 °C with an EASY-Spray emitter (Thermo, ES991). Mobile phase A was 0.1% formic acid and mobile phase B was 80% acetonitrile, 0.1% formic acid. The gradient separation method at flow rate 300 nl min[−1] was as follows: for 80 min gradient from 5–35% B, for 5 min up to 95% B, for 5 min isocratic at 95% B, re-equilibration to 5% B in 5 min, for 5 min isocratic at 5% B. Precursors between 380 and

1,400 *m/z* and charge states 3–8 were selected at 120,000 resolution in the top speed mode in 3 s and were isolated for stepped HCD fragmentation (collision energies (%) = 21, 27, 34) with quadrupole isolation width 1.6 Th, Orbitrap detection with 30,000 resolution and 70 ms maximum injection time. Targeted mass spectrometry precursors were dynamically excluded from further isolation and activation for 45 s with 10 ppm mass tolerance. Identification of crosslinked peptides was performed in Proteome Discoverer 2.4 (Thermo) with the Xlinkx search engine in the MS2 mode for DSSO/+158.004 Da (K). Precursor and fragment mass tolerances were 10 ppm and 0.02 Da respectively with maximum 2 trypsin missed cleavages allowed. Carbamidomethyl at C was selected as static modification. Spectra were searched against a UniProt FASTA file containing Homo sapiens reviewed entries. Crosslinked peptides were filtered at FDR < 0.01 using the percolator node and target–decoy database search.

### Statistics and reproducibility

Three biological replicates were performed for metaphase spreads, except for those in Extended Data Fig. 1h,i, which were repeated twice. All other experiments were independently replicated a minimum of three times except for Fig. 3c and Extended Data Figs. 2f,g and 7c which were repeated twice. All attempts at replication were successful.

### Reporting summary

Further information on research design is available in the Nature Portfolio Reporting Summary linked to this article.

### Data availability

The cryo-EM composite map of the RAP1–DNA-PK complex was deposited to the Electron Microscopy Data Bank under accession code EMD-19065. Corresponding atomic coordinates were deposited to the Protein Data Bank under PDB ID 8RD4. Constituent cryo-EM maps for locally refined KU–RAP1 and DNA-PKcs regions as well as a consensus map of the full complex were deposited under accession codes EMD-19249, EMD-19252 and EMD-19245 respectively. Mass spectrometry proteomics data have been deposited to the ProteomeXchange Consortium via the PRIDE[60] partner repository with the dataset identifier PXD047643.

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

**Acknowledgements** The authors thank P. Martin, R. Broderick, A. Radzisheuskaya and H. Mischo for reagents and advice and G. Coster and T. de Lange for comments on the manuscript. M.E.D. is funded by a Cancer Research UK Career Development Award (C68409/A28129). F.L. is supported by grants from Cancerfonden (21 1732 Pj), Vetenskapsrådet (2021-02788) and Knut and Alice Wallenberg Foundation. J.M. is funded by a Cancer Research UK Senior Cancer Research Fellowship (RCCSCF-Nov22/100001). S.G. is supported by a Wellcome Trust Investigator Award (214311/Z/18/Z) and a Lister Institute Research Prize Fellowship.

**Author contributions** M.E.D. conceived the study, with genetic experiments devised in collaboration with F.L. MEF experiments were performed by C.S. with help from A.d.S. RPE-1 cell experiments were performed by M.E.D. and P.E. with advice from J.M. Experiments in Figs. 2, 3 and 5 were performed by P.E. with help from C.E.L.F. For Fig. 2b, S.G. designed the expression construct for shelterin, which was purified by O.I. Samples for Fig. 4 were prepared by P.E. and C.E.L.F. O.I prepared and screened grids and P.E and C.E.L.F. performed data processing. T.I.R. performed crosslinking mass spectrometry for Extended Data Fig. 5, on a sample prepared by P.E. Supervision was provided by J.S.C., S.G., F.L. and M.E.D. The manuscript was written by M.E.D. with input from all authors.

**Competing interests** The authors declare no competing interests.

**Additional information**
**Correspondence and requests for materials** should be addressed to Francisca Lottersberger or Max E. Douglas.

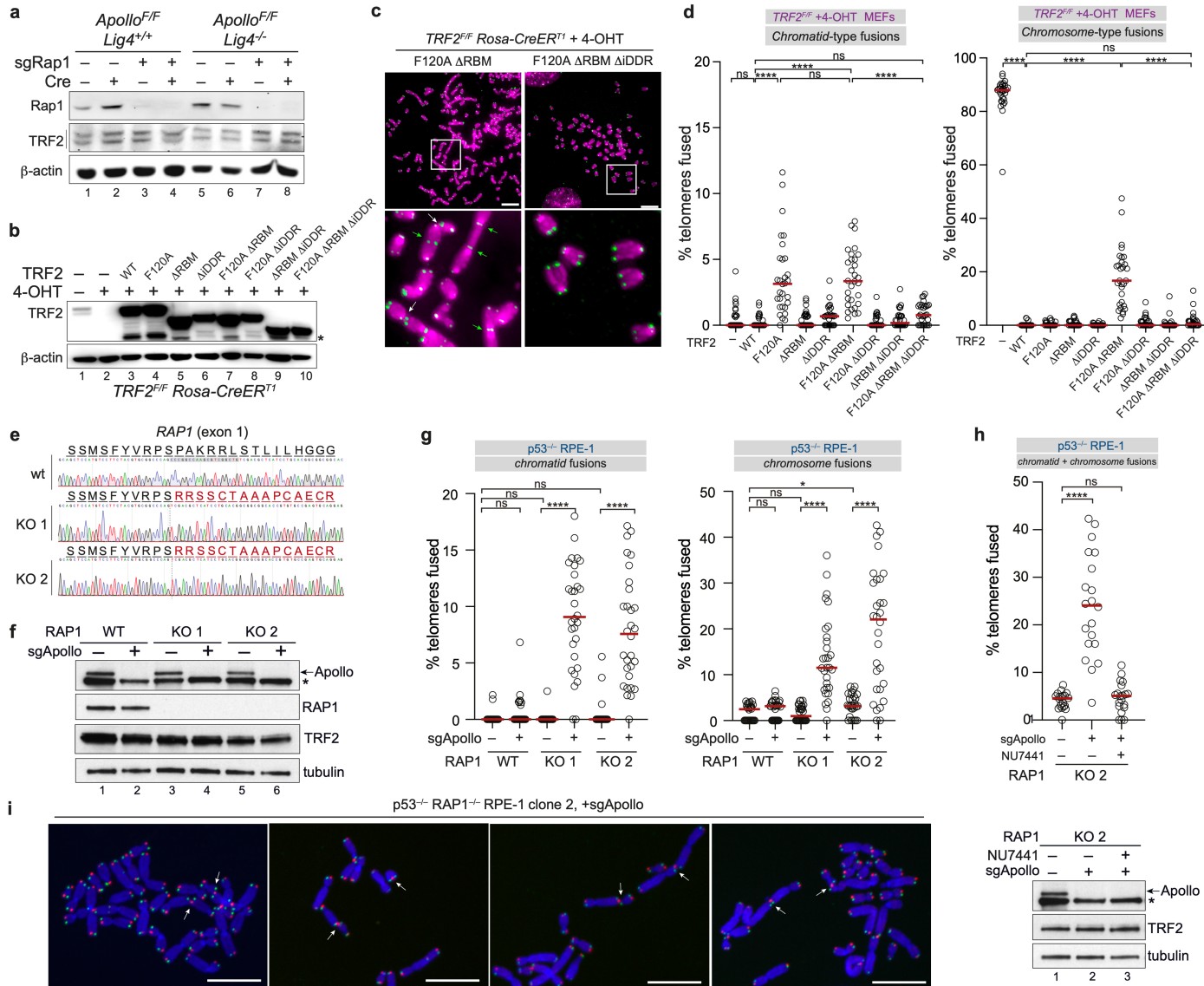

**Extended Data Fig. 1 | RAP1 and APOLLO redundantly prevent cNHEJ at telomeres in mouse and human cells. a**. Immunoblot showing effective removal of Rap1 and persistence of TRF2 as indicated after Crispr-mediated Rap1 deletion and/or Cre-mediated deletion of Apollo in ApolloF/F Lig4⁺/⁺ or ApolloF/F Lig4⁻/⁻ MEFs. Beta-actin as loading control. **b**. Immunoblot showing expression of TRF2 in SV40LT-immortalized TRF2F/F RsCre-ERT1 MEFs transduced with an empty vector control (-), mouse TRF2 WT or TRF2 alleles habouring a F120A mutation and/or deleted of the RAP1 binding motif (ΔRBM) and/or deleted of the DNA-damage response motif (ΔiDDR) 120 h after deletion of endogenous TRF2 with 4-OHT. Beta-actin as loading control. **c**. Representative FISH of metaphase spreads from cells as described in b. Telomeres detected with Cy3-(TTAGGG)3 (green). DNA stained with DAPI (magenta). White and green arrows highlight chromatid- and chromosome type fusions respectively. Scale bar = 10 μm **d**. Quantification of percentage telomeres involved in chromatid and chromosome fusions per metaphase after deletion of endogenous TRF2 in MEFs expressing the indicated TRF2 mutants, as described in b. Data from 3 independent experiments, 10 metaphases per experiment

(n = 30 total), with median. Statistics by ordinary One-way ANOVA. 'ns' not significant, ****P ≤ 0.0001. **e**. Sequencing of the RAP1 locus in p53⁻/⁻ RAP1⁺/⁺ and p53⁻/⁻ RAP1⁻/⁻ RPE-1 cells. 19 bp deletion in knockout clones generates an altered reading frame terminating in a stop codon after a total of 135 amino acids. **f**. Immunoblot showing Apollo protein levels 120 h after transduction of p53⁻/⁻ RAP1⁺/⁺ and p53⁻/⁻ RAP1⁻/⁻ RPE−1 cells with Cas9 and sgApollo or sgControl. Alpha-tubulin as loading control. **g**. Chromatid- and chromosome-type telomere fusions quantified after Apollo deletion as in (f). From three independent experiments, 10 metaphases from each, with median. Statistics as in (d). **h**. Upper panel as in (g) but cells treated with 2 μM DNA-PK inhibitor NU7741 for 48 h prior to cell collection. From two independent experiments, 10 metaphases each, with median. Statistics as in (d). Lower panel as in (f). **i**. Representative coFISH images of RAP1 KO clone 2 120 h after transduction with Cas9 and sgApollo. Telomeres detected with Cy3-(TTAGGG)3 (red = leading) and FITC-(CCCTTA)3 (green = lagging). Arrows highlight leading:leading telomere fusions. For gel source data see Supplementary Fig. 1.

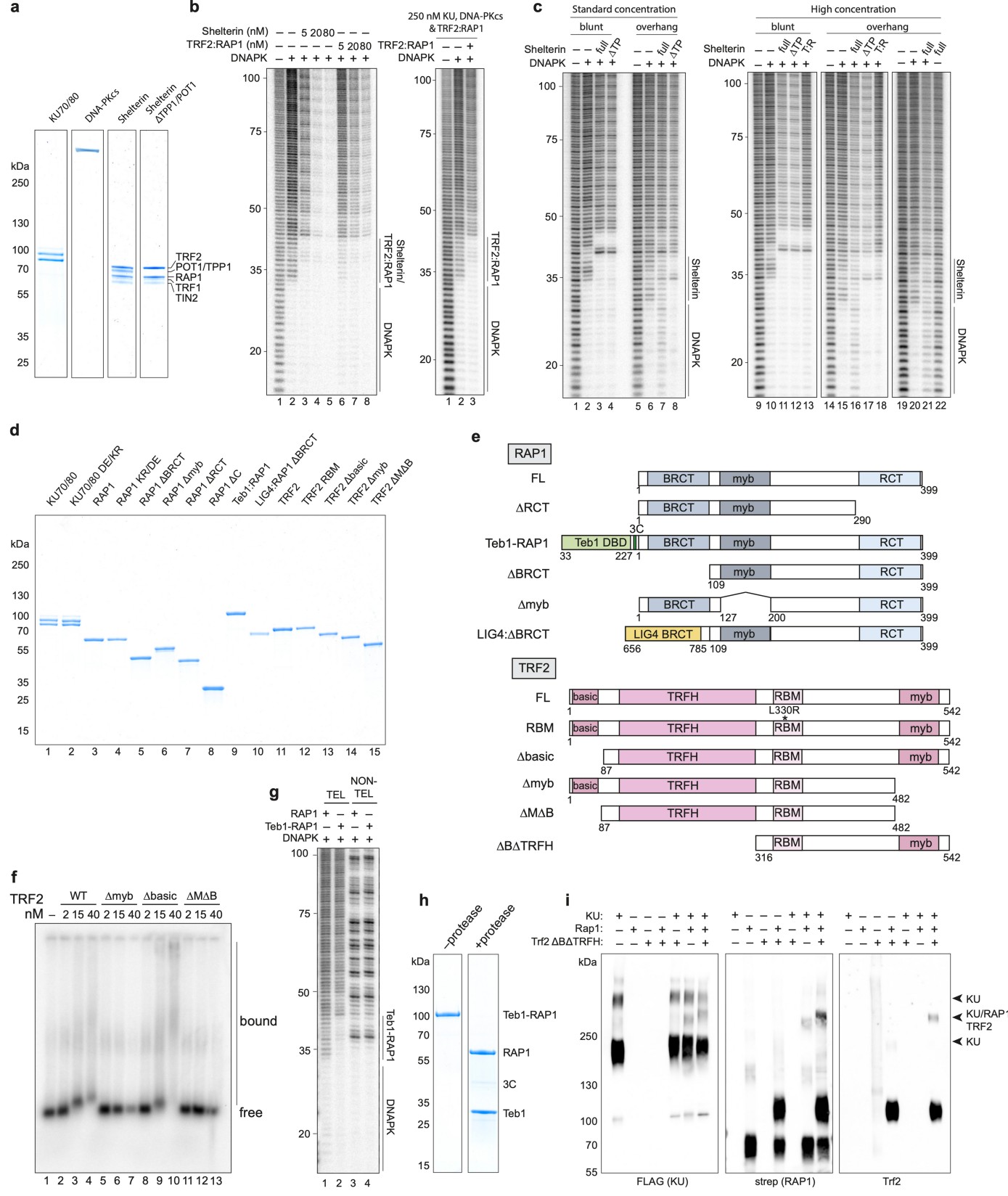

**Extended Data Fig. 2** | See next page for caption.

**Extended Data Fig. 2 | Deletion constructs, purified proteins and data relating to Figs. 2 and 3. a**. Purified proteins as indicated, separated on a denaturing tris-glycine polyacrylamide gel and stained with Instant Blue. **b**. DNase I footprinting performed as in Fig. 2a with shelterin or TRF2/RAP1 at increasing concentrations (left panel) or with KU70/80, DNA-PKcs and TRF2/RAP1 each at 250 nM (right panel). Template concentration increased to 20 nM for the experiment on the right. Nucleotides from the 5′ telomeric end indicated. **c**. DNase I footprinting performed as in Fig. 2a, except shelterin was added coincident with KU70/80 and DNA-PKcs. Lanes 9-22 contained 8 nM template, 100 nM KU/DNA-PKcs and 40 nM shelterin or TRF2:RAP1 as indicated. Cleavage within the 10 bp footprint derives from a change in template sequence register compared with the standard template in Figs. 2–4. Nucleotides from the 5′ telomeric end indicated. **d**. Purified proteins as indicated, separated on a denaturing tris-glycine polyacrylamide gel and stained with Instant Blue. **e**. Domain organisation of RAP1 and TRF2 mutants. TRF2 L330R (equivalent to L288R in the shorter TRF2 isoform) prevents binding to RAP1 (Chen et al, 2011). PreScission cleavage site highlighted by 3 C. Also see Supplementary Information Table 2. **f**. Electrophoretic mobility shift assay of telomeric DNA bound by TRF2. **g**. DNase I footprinting performed as in Fig. 3d with proteins omitted as indicated. Nucleotides from the 5′ telomeric end indicated. **h**. Cleavage of Teb1 DBD from RAP1 using PreScission protease. Proteins were separated on a denaturing tris-glycine polyacrylamide gel and stained with Instant Blue. **i**. Protein cross-linking analysis of KU, RAP1 and TRF2 containing only RBM and myb domains, in the presence of DNA. Arrowheads mark the position of cross-linked species containing only KU, or KU, RAP and TRF2. For gel source data see Supplementary Fig. 1.

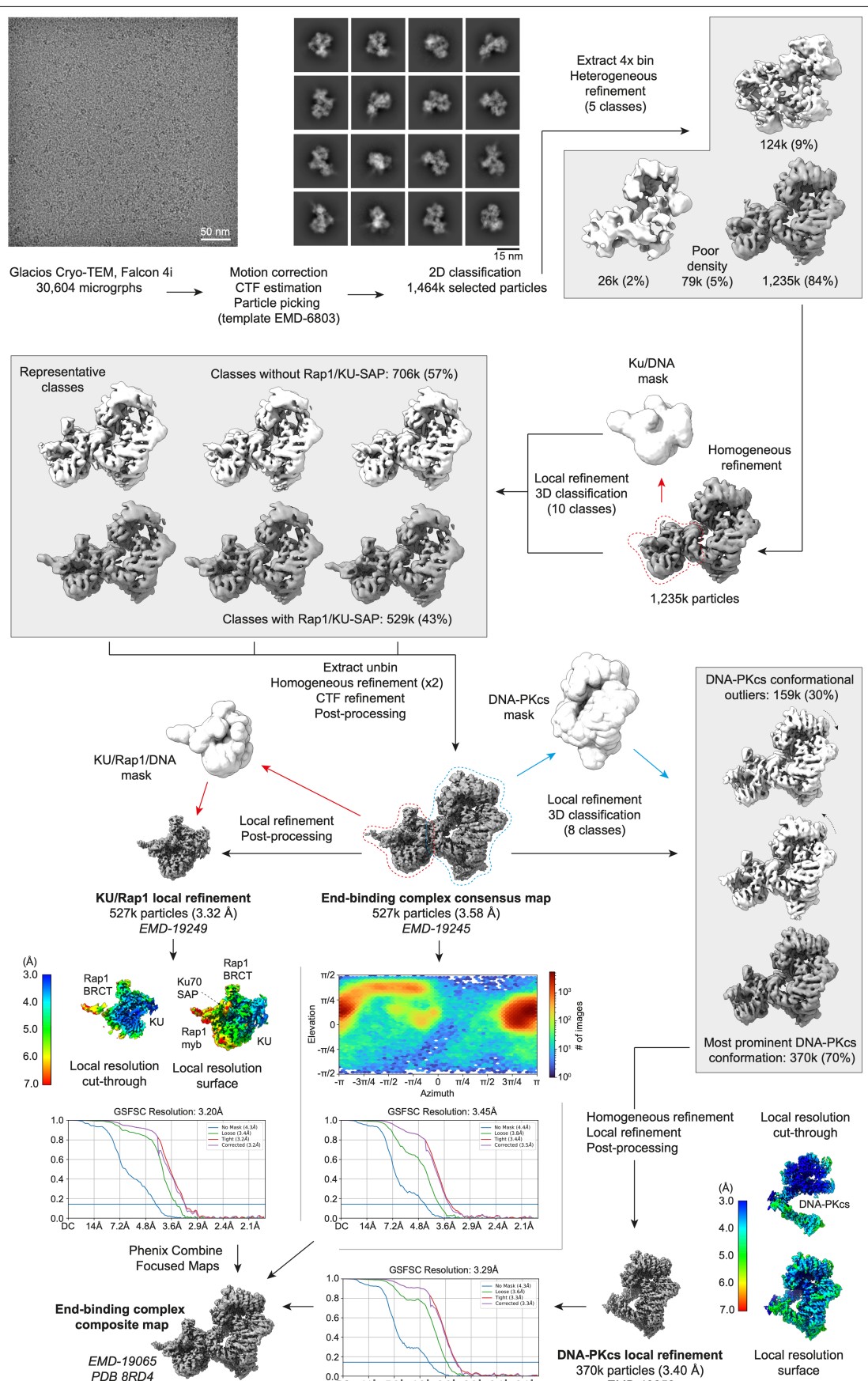

**Extended Data Fig. 3 | Cryo-EM data processing pipeline for the RAP1:DNA-PK complex on DNA.** Schematic showing the cryoSPARC classification and refinement steps used to obtain the RAP1:DNA-PK structure. Also see Extended Data Table 1.

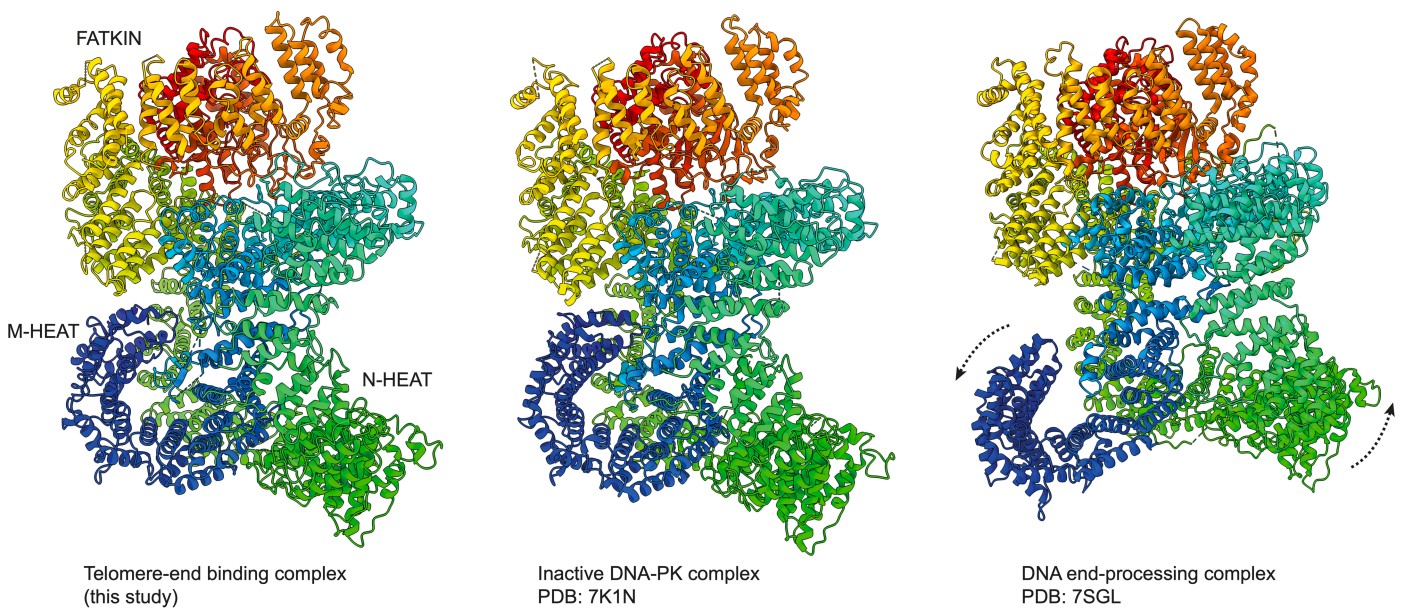

Telomere-end binding complex
(this study)

Inactive DNA-PK complex
PDB: 7K1N

DNA end-processing complex
PDB: 7SGL

**Extended Data Fig. 4 | DNA-PKcs conformation in the RAP1:DNA-PK complex.** DNA-PKcs models from the structures indicated, showing the M-HEAT and N-HEAT domains adopting an 'inactive' conformation in the telomere end binding complex.

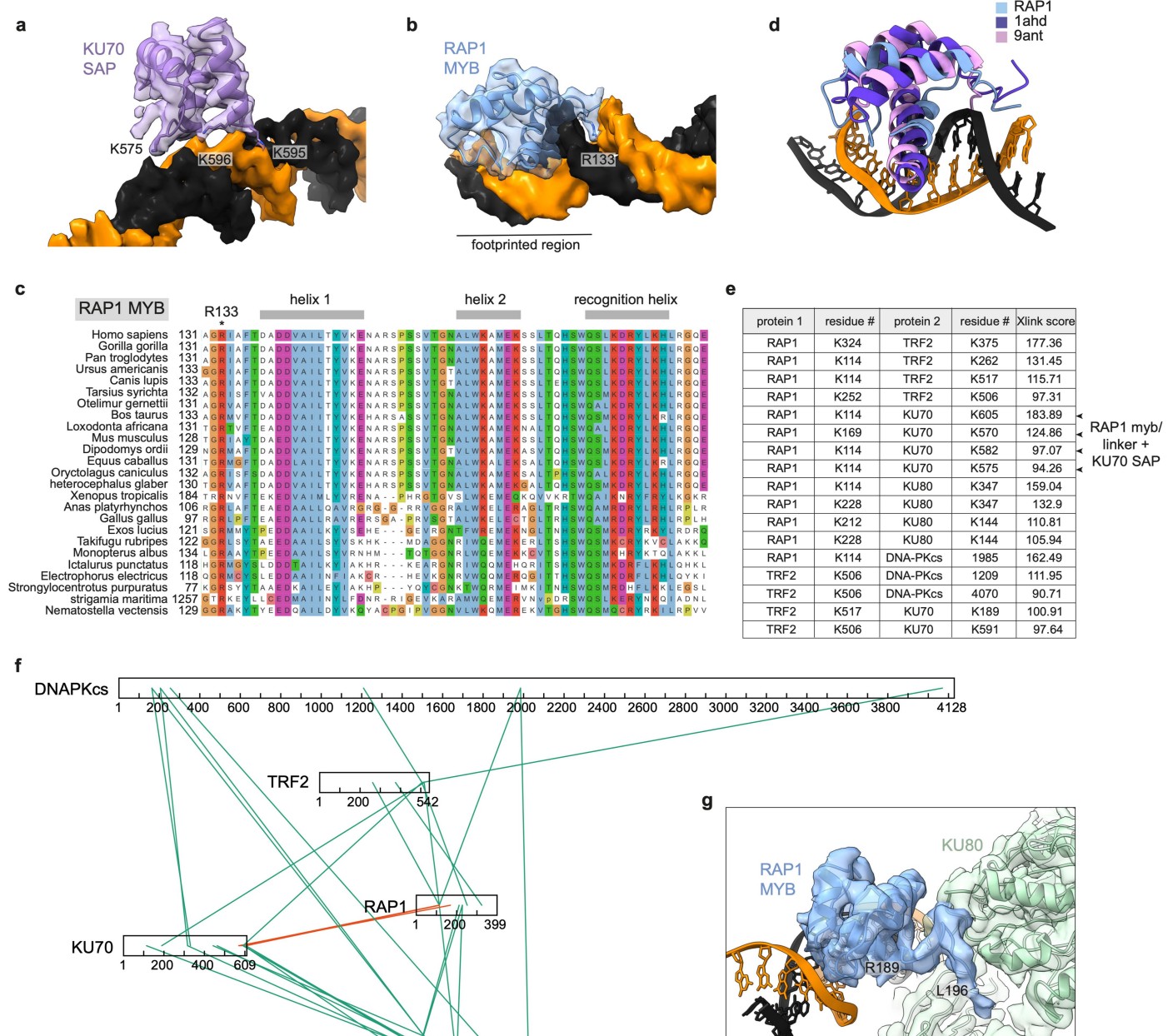

**Extended Data Fig. 5 | Binding of the KU70 SAP and RAP1 myb domains to DNA. a**. Density map and model of the KU70 SAP:DNA interaction extracted from the complete structure, showing K575, K595 and K596 coordinating the phosphate backbone **b**. Density map and model of the RAP1 myb:DNA interaction extracted from the complete structure, showing the recognition helix sitting in the major groove, and R133 acting as an N-terminal arm inserted into the neighbouring minor groove. Region protected from DNase I indicated **c**. Sequence alignment of the human RAP1 myb domain. Alignment adopts Clustal X colouring **d**. Comparison of the DNA-bound human RAP1 myb domain (blue) with homeotic protein antennapedia (purple - 1ahd) and HOX-B1 (pink - 1b72). Structures were aligned via the DNA. **e**. Table of intermolecular chemical crosslinks detected by XLMS. Data were thresholded with an XlinkX score ≥90 and crosslinks between KU and DNA-PKcs were excluded. **f**. Intermolecular chemical crosslinks detected by XLMS. Data were thresholded with an XlinkX score ≥90. **g**. Density map and model of the RAP1 loop C-terminal to the myb domain anchored to KU80.

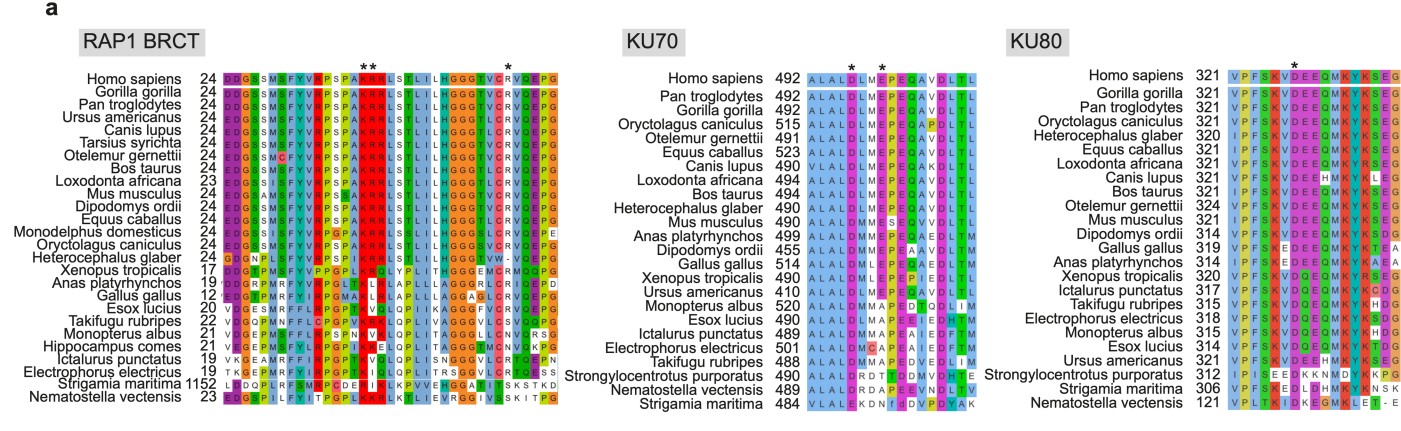

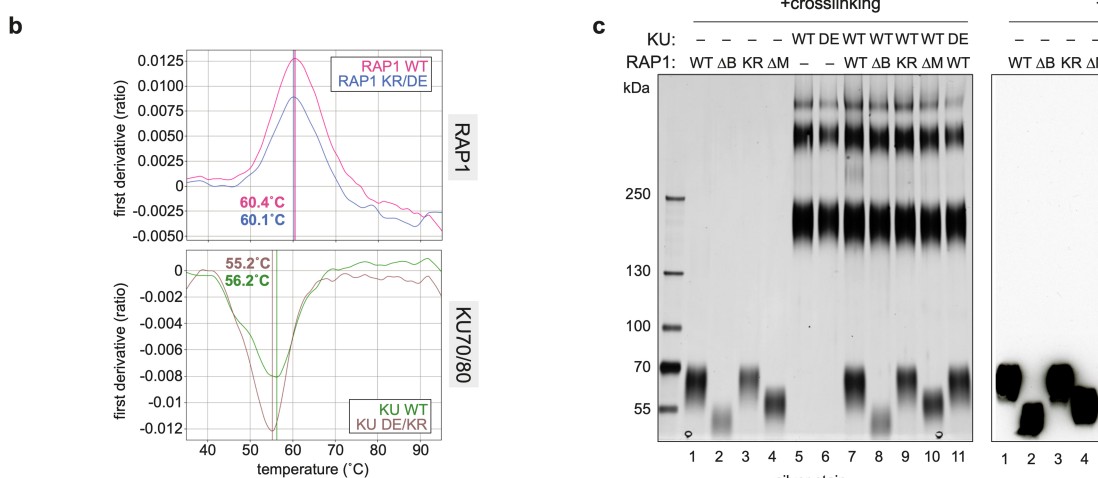

**Extended Data Fig. 6 | Binding of the RAP1 BRCT domain to KU70 vWA.**
**a.** Sequence alignment of the RAP1 (left), KU70 (middle) and KU80 (right) regions that engage in the BRCT:KU interaction. Asterisks mark K39, R40 and R55 in RAP1, which are changed to aspartate or glutamate in the KR/DE mutant. Also, D496 and E499 in KU70 and D326 in KU80, which are changed to lysine or arginine in the KU DE/KR mutant. Alignment adopts Clustal X colouring **b.** Nano-scale differential scanning fluorimetry analysis of the RAP1 BRCT or KU mutants as indicated, showing no significant effect of the point mutations on protein folding. **c.** Protein cross-linking analysis of RAP1 and KU in the presence of DNA. Proteins were mixed with crosslinker and reaction products were separated on a denaturing tris-acetate polyacrylamide gel and analysed by silver staining or immunoblotting as indicated. ΔB and ΔM mark Rap1 BRCT and myb domain deletion mutants respectively. RAP1 KR/DE (KR) contains K39D, R40E and R55E mutations. KU DE/KR (DE) contains KU70 D496K, E499R and KU80 D327K mutations. For gel source data see Supplementary Fig. 1.

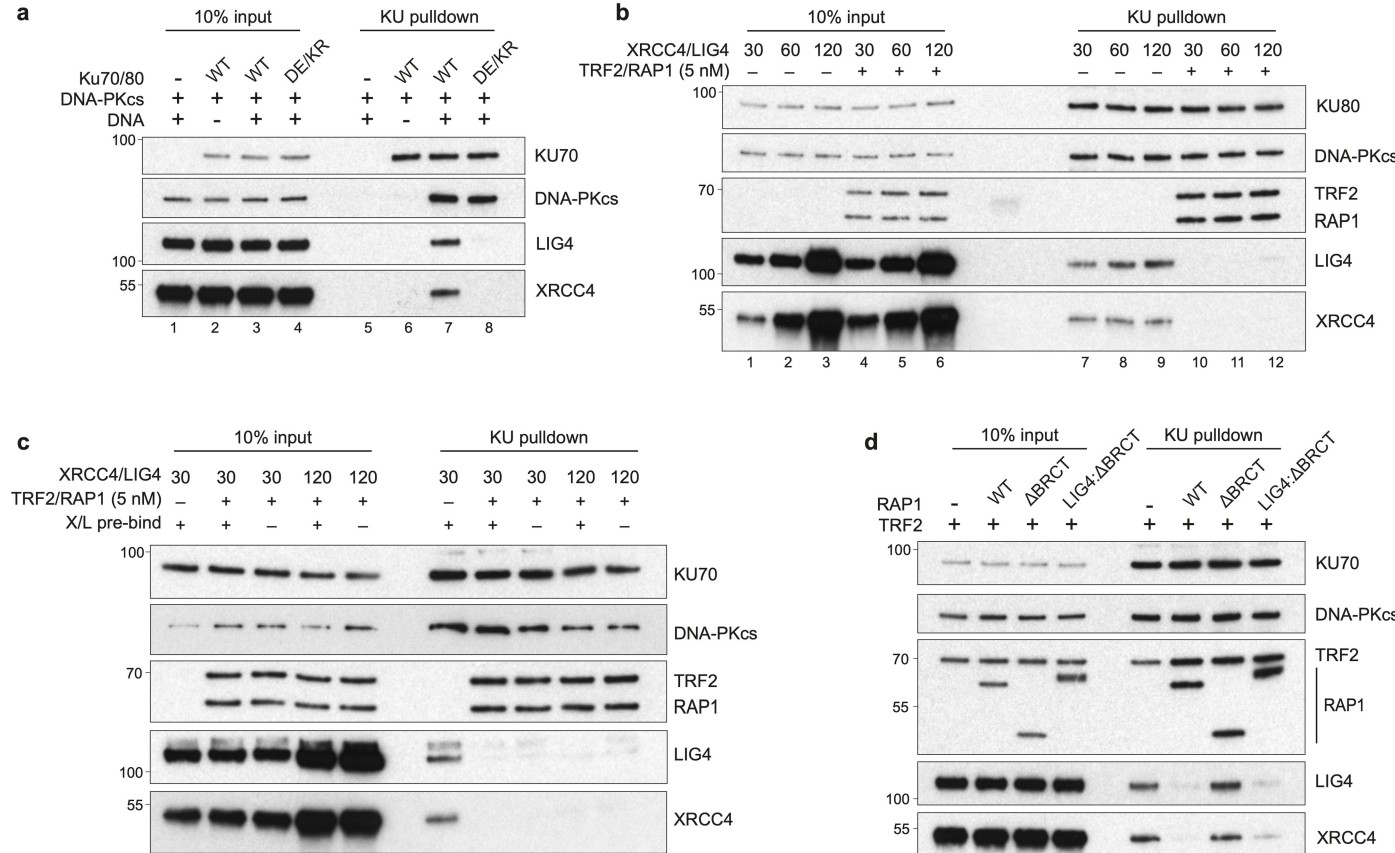

**Extended Data Fig. 7 | Supplementary DNA-PK binding assays related to Fig. 5. a**. KU pulldown experiment without TRF2 or RAP1, executed as described in Fig. 5a. KU DE/KR contains the mutations KU70 D496K, E499R and KU80 D327K **b**. As in (a), but with reactions containing 5 nM TRF2/RAP1 complex and increasing concentrations of XRCC4/LIG4 as indicated. **c**. As in (b). 'X/L pre-bind' indicates reactions where XRCC4/LIG4 was incubated with KU70/80, DNA-PKcs and DNA template for 5 min prior to the addition of TRF2/RAP1. **d**. As in (b), but with 30 nM XRCC4/LIG4, and the RAP1 proteins indicated. For gel source data see Supplementary Fig. 1.

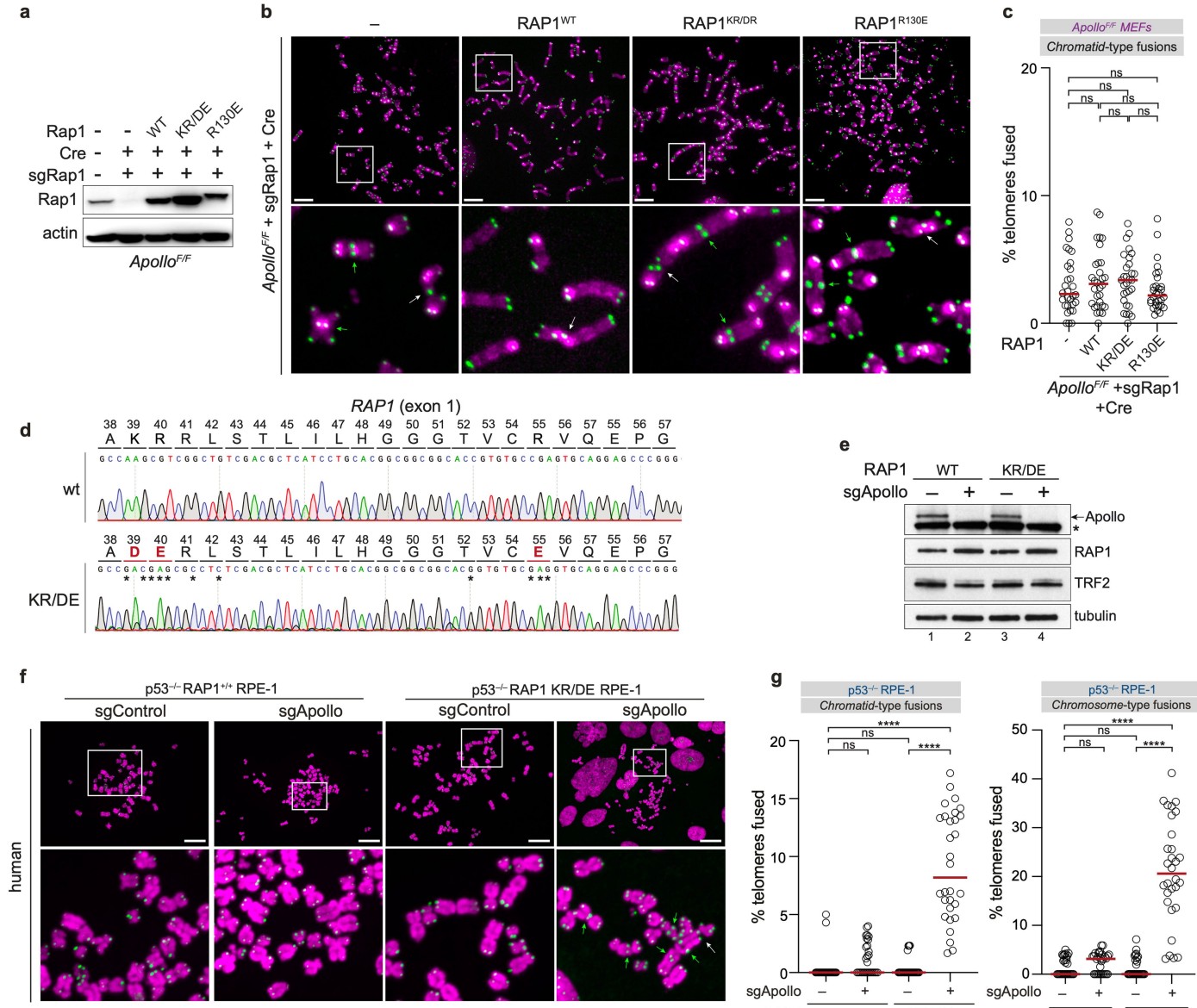

**Extended Data Fig. 8 | Mutation analysis of RAP1 in mouse and human cells.**
**a.** Immunoblot showing over-expression of mouse RAP1, RAP1 KR/DE and RAP1 R130E after Crispr- and Cre-mediated deletion of Rap1 and Apollo respectively in ApolloF/F Lig4+/+ MEFs. Beta-actin as loading control. **b.** Representative FISH of metaphase spreads of ApolloF/F Lig4+/+ MEFs expressing the indicated RAP1 mutants or a control empty vector (EV) 96-120 h after deletion of endogenous Rap1 with sgRNA and 120 h after deletion of APOLLO with Hit & Run Cre. Telomeres detected with Cy3-(TTAGGG)3 (green). DNA stained with DAPI (magenta). White and green arrows highlight chromatid- and chromosome type fusions respectively. Scale bar = 10 μm **c.** Quantification of percentage telomeres involved in chromatid fusions per metaphase after expression of Rap1 or the empty vector control (-) and removal of endogenous Rap1 and Apollo as described in b. Data from 3 independent experiments, 10 metaphases per experiment (n = 30 total), with median. Statistics by ordinary One-way ANOVA. 'ns' not significant. **d.** Sequence analysis of the human RAP1 locus in RAP1 WT and RAP1 KR/DE p53−/− RPE-1 cells. Targeted mutations in KR/DE are marked with an asterisk, with altered amino acids highlighted in red **e.** Immunoblot showing Apollo, RAP1 and TRF2 levels after Crispr-mediated deletion in p53−/− RAP1 WT and p53−/− RAP1 KR/DE RPE-1 cells. Asterisk marks a non-specific band detected by the anti-Apollo antibody. Alpha-tubulin as loading control. **f.** Representative FISH of metaphase spreads of RAP1 WT and RAP1 KR/DE p53−/− RPE-1 cells 120 h after transduction with Cas9 and sgApollo or sgControl as indicated. Telomeres detected with Cy3-(TTAGGG)3 (green). DNA stained with DAPI (magenta). White and green arrows highlight chromatid- and chromosome type fusions respectively. **g.** Quantification of the percentage of telomeres involved in chromatid and chromosome fusions per metaphase after removal of Apollo in RAP1 WT and RAP1 KR/DE p53−/− RPE-1 cells as indicated. Data from 3 independent experiments, 10 metaphases per experiment (n = 30 total), with median. Statistics by ordinary One-way ANOVA. 'ns' not significant, ****P ≤ 0.0001. For gel source data see Supplementary Fig. 1.

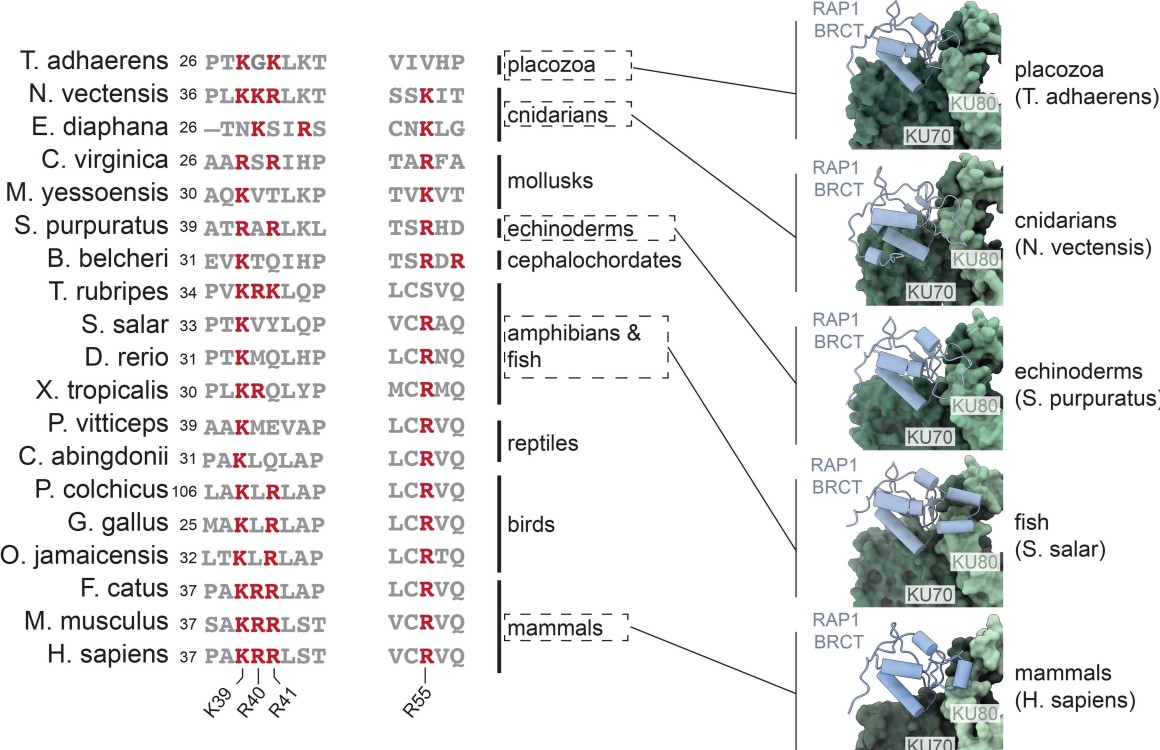

**Extended Data Fig. 9 | Conservation analysis of the RAP1 BRCT:KU interaction.** Amino acid sequences of RAP1 from the metazoan species indicated were aligned using the Muscle algorithm. The basic patch composed of K39, R40, R41 and R55 in the BRCT domain is shown with basic residues displayed in red. Corresponding structure predictions for the RAP1:KU complex made using AlphaFold 3 are shown for the species indicated, revealing that the BRCT:KU interaction is likely to occur widely across metazoa.

**Extended Data Table 1 | Cryo-EM data collection, refinement and validation statistics**

| | End-binding complex consensus map | KU:RAP1 local refinement | DNA-PKcs local refinement | End-binding complex composite map* |
|---|---|---|---|---|
| Accession codes | EMD-19245 | EMD-19249 | EMD-19252 | EMD-19065 PDB 8RD4 |
| **Data collection and processing** | | | | |
| Magnification | x150,000 | x150,000 | x150,000 | |
| Voltage (kV) | 200 | 200 | 200 | |
| Electron exposure (e–/Å$^2$) | 50.0 | 50.0 | 50.0 | |
| Defocus range (μm) | -1.0 to -2.5 | -1.0 to -2.5 | -1.0 to -2.5 | |
| Pixel size (Å) | 0.94 | 0.94 | 0.94 | |
| Symmetry imposed | C1 | C1 | C1 | |
| Initial particle images (no.) | 7,463,361 | 7,463,361 | 7,463,361 | |
| Final particle images (no.) | 526,885 | 526,885 | 370,172 | |
| Map resolution (Å) | 3.58 | 3.32 | 3.40 | |
| FSC threshold | 0.143 | 0.143 | 0.143 | |
| Map resolution range (Å) | 3.0-7.0 | 3.0-7.0 | 3.0-6.0 | |
| **Refinement (statistics from MolProbity)** | | | | |
| Initial model used (PDB code) | | | | 7K1K, 1FEX, 1JJR |
| Model resolution (Å) | | | | 3.7 |
| FSC threshold | | | | 0.5 |
| Model resolution range (Å) | | | | 2.9-4.0 |
| Map sharpening *B* factor (Å$^2$) | | | | -70 |
| Model composition | | | | |
| Non-hydrogen atoms | | | | 40,731 |
| Protein residues | | | | 4,868 |
| Ligands | | | | 0 |
| *B* factors (Å$^2$) | | | | |
| Protein | | | | -75.75 |
| Ligand | | | | - |
| R.m.s. deviations | | | | |
| Bond lengths (Å) | | | | 0.004 |
| Bond angles (°) | | | | 0.711 |
| Validation | | | | |
| MolProbity score | | | | 2.17 |
| Clashscore | | | | 12.66 |
| Poor rotamers (%) | | | | 0.46 |
| Ramachandran plot | | | | |
| Favored (%) | | | | 89.85 |
| Allowed (%) | | | | 10.13 |
| Disallowed (%) | | | | 0.02 |
| Q-score | | | | 0.4280 |

* Composite map of consensus volume (EMD-19245) and local refinements (EMD-19249, EMD-19252)

# Reporting Summary

## Statistics

For all statistical analyses, confirm that the following items are present in the figure legend, table legend, main text, or Methods section.

| n/a | Confirmed | |
|---|---|---|
| ☐ | ☒ | The exact sample size (*n*) for each experimental group/condition, given as a discrete number and unit of measurement |
| ☐ | ☒ | A statement on whether measurements were taken from distinct samples or whether the same sample was measured repeatedly |
| ☐ | ☒ | The statistical test(s) used AND whether they are one- or two-sided<br>*Only common tests should be described solely by name; describe more complex techniques in the Methods section.* |
| ☒ | ☐ | A description of all covariates tested |
| ☒ | ☐ | A description of any assumptions or corrections, such as tests of normality and adjustment for multiple comparisons |
| ☐ | ☒ | A full description of the statistical parameters including central tendency (e.g. means) or other basic estimates (e.g. regression coefficient) AND variation (e.g. standard deviation) or associated estimates of uncertainty (e.g. confidence intervals) |
| ☐ | ☒ | For null hypothesis testing, the test statistic (e.g. *F*, *t*, *r*) with confidence intervals, effect sizes, degrees of freedom and *P* value noted<br>*Give P values as exact values whenever suitable.* |
| ☒ | ☐ | For Bayesian analysis, information on the choice of priors and Markov chain Monte Carlo settings |
| ☒ | ☐ | For hierarchical and complex designs, identification of the appropriate level for tests and full reporting of outcomes |
| ☒ | ☐ | Estimates of effect sizes (e.g. Cohen's *d*, Pearson's *r*), indicating how they were calculated |

*Our web collection on statistics for biologists contains articles on many of the points above.*

## Software and code

Policy information about availability of computer code

| Data collection | Cryo-EM data collection: ThermoFisher EPU v3.2<br>Phosphor imaging: Amersham TYPHOON Scanner Control Software v3.0.0.2<br>Nano differential scanning fluorimetry data collection: Tycho NT.6 v1.3.2.880<br>Microscopy: Slide book (3i), DeltaVision RT microscope system (GE Healthcare)<br>Gel and membrane scanning: ChemiDoc (Bio-Rad) |
|---|---|
| Data analysis | Structural data analysis: cryoSPARC v4.3.1, Phenix v1.20.1, Coot v0.9.8.92, Namdinator, UCSF ChimeraX v1.6.1, Alphafold 3<br>Nano differential scanning fluorimetry data analysis: Tycho NT.6 v1.3.2.880<br>Protein sequence alignment analysis: ProViz (online tool)<br>Image processing: ImageJ2 (FIJI) v2.14.0, FIJI v1.0, Adobe Photoshop v25.0.0, Adobe Illustrator v25.0.0<br>Metaphase spread graphs: GraphPad Prism (v9 or 10.0.0) |

For manuscripts utilizing custom algorithms or software that are central to the research but not yet described in published literature, software must be made available to editors and reviewers. We strongly encourage code deposition in a community repository (e.g. GitHub). See the Nature Portfolio guidelines for submitting code & software for further information.

## Data

Policy information about availability of data

All manuscripts must include a data availability statement. This statement should provide the following information, where applicable:

- Accession codes, unique identifiers, or web links for publicly available datasets
- A description of any restrictions on data availability
- For clinical datasets or third party data, please ensure that the statement adheres to our policy

The cryo-EM composite map of the RAP1:DNA-PK complex was deposited to the Electron Microscopy Data Bank under accession code EMD-19065. Corresponding atomic coordinates were deposited to the Protein Data Bank under PDB ID 8RD4. Constituent cryo-EM maps for locally refined KU:RAP1 and DNA-PKcs regions as well as a consensus map of the full complex were deposited under accession codes EMD-19249, EMD-19252 and EMD-19245 respectively. Mass spectrometry proteomics data have been deposited to the ProteomeXchange Consortium via the PRIDE59 partner repository with the dataset identifier PXD047643.

## Research involving human participants, their data, or biological material

Policy information about studies with human participants or human data. See also policy information about sex, gender (identity/presentation), and sexual orientation and race, ethnicity and racism.

| | |
|---|---|
| Reporting on sex and gender | N/A |
| Reporting on race, ethnicity, or other socially relevant groupings | N/A |
| Population characteristics | N/A |
| Recruitment | N/A |
| Ethics oversight | N/A |

Note that full information on the approval of the study protocol must also be provided in the manuscript.

# Field-specific reporting

Please select the one below that is the best fit for your research. If you are not sure, read the appropriate sections before making your selection.

☒ Life sciences          ☐ Behavioural & social sciences          ☐ Ecological, evolutionary & environmental sciences

For a reference copy of the document with all sections, see nature.com/documents/nr-reporting-summary-flat.pdf

# Life sciences study design

All studies must disclose on these points even when the disclosure is negative.

| | |
|---|---|
| Sample size | No statistical method was used to predetermine sample size. For chromosome fusion experiments, sample size was determined based on previous similar experiments: n=30-40 metaphases over 3-4 independent experiments (Lottersberger et al., Cell, 2015). For cryo-EM data collection, sample size was determined to obtain sufficient resolution for model building. |
| Data exclusions | For chromosome fusion experiments, metaphases where it was not possible to perform robust scoring (i.e. less than 10 chromosomes, too many overlapping chromosomes) were excluded. For cryo-EM image processing, particles that did not align into discernable 2D or 3D classes were excluded. No other data were excluded from experiments in the study. |
| Replication | All analyses were independently replicated a minimum of 3 times except experiments in Figure 3c as well as Extended Data Fig 1h,i; 2f,g and 7c which were replicated twice. All attempts at replication were successful. |
| Randomization | For cryo-EM volumes, final particle subsets were randomly split into two independent halves to calculate Fourier Shell Correlation. For other experiments, no randomization was applied. Appropriate controls were included where applicable. |
| Blinding | Mouse cell metaphase spreads were not scored blind as sample identity was obvious to the experienced investigator. Human metaphase spreads were scored blind. Blinding was not compatible with other experiments, which involved electrophoresis. |

# Reporting for specific materials, systems and methods

We require information from authors about some types of materials, experimental systems and methods used in many studies. Here, indicate whether each material, system or method listed is relevant to your study. If you are not sure if a list item applies to your research, read the appropriate section before selecting a response.

## Materials & experimental systems

| n/a | Involved in the study |
|---|---|
| ☐ | ☒ Antibodies |
| ☐ | ☒ Eukaryotic cell lines |
| ☒ | ☐ Palaeontology and archaeology |
| ☒ | ☐ Animals and other organisms |
| ☒ | ☐ Clinical data |
| ☒ | ☐ Dual use research of concern |
| ☒ | ☐ Plants |

## Methods

| n/a | Involved in the study |
|---|---|
| ☒ | ☐ ChIP-seq |
| ☒ | ☐ Flow cytometry |
| ☒ | ☐ MRI-based neuroimaging |

# Antibodies

| | |
|---|---|
| Antibodies used | Mouse monoclonal anti-DNA-PKcs, Invitrogen MA5-13238, clone 18-2, 1:100 dilution<br>Rabbit-polyclonal anti-RAP1, Bethyl A300-306A, 1:4000 dilution<br>Rabbit monoclonal anti-RAP1, Cell Signalling 5433, clone D9H4, 1:1000 dilution<br>Rabbit monoclonal anti-TRF2, Cell Signalling 13136, clone D1Y5D, 1:1000 dilution<br>Rabbit polyclonal anti-DCLRE1B, Atlas HPA064934, 1:100 dilution<br>Mouse monoclonal anti-XRCC4, Santa Cruz sc-271087, clone C-4, 1:500 dilution<br>Mouse monoclonal anti-β-actin, Cell Signalling 3700, clone 8H10D10, 1:4000 dilution<br>Mouse monoclonal anti-α-tubulin, Sigma T9026, clone DM1A, 1:1000 dilution<br>Mouse monoclonal anti-FLAG, Sigma F1804, clone M2, 1:1000 dilution<br>Rabbit polyclonal anti-Strep-tag II, Abcam ab76949, 1:1000 dilution<br>Goat polyclonal anti-Rabbit IgG HRP, Cell Signalling 7074<br>Horse anti-Mouse IgG HRP, Cell Signalling 7076<br>Donkey polyclonal anti-Rabbit IgG HRP, Cytiva NA934V<br>Goat polyclonal anti-Rabbit IgG HRP, Invitrogen 31460<br>Goat polyclonal anti-Mouse IgG HRP, Invitrogen 31430 |
| Validation | Antibodies for mouse RAP1 (Cell Signalling 5433), mouse TRF2 (Cell Signalling 13136), human Apollo (Atlas HPA064934) and human RAP1 (Bethyl A300-306A) were validated in this study by Crispr-mediated knockout of target genes followed by western blotting, confirming loss of the relevant band. Other primary antibodies have been previously validated as follows:<br><br>anti-DNA-PKcs, Invitrogen MA5-13238 - validated by a strong band of the correct molecular weight in human LS174T cells<br>anti-XRCC4, Santa Cruz sc-271087 - validated by Western blotting against purified recombinant XRCC4 protein<br>anti-β-actin, Cell Signalling 3700 - validated by Western blotting against recombinant β-actin protein<br>anti-α-tubulin, Sigma T9026 - validated by a strong band of the correct molecular weight in mouse cell lysates and a characteristic tubulin immunofluorescence pattern in fixed human cells.<br>Anti-FLAG and anti-strep tag antibodies - validated by detection of purified recombinant FLAG or strep-tagged proteins |

# Eukaryotic cell lines

Policy information about cell lines and Sex and Gender in Research

| | |
|---|---|
| Cell line source(s) | SV40-LT ApolloF/F Lig4+/+, ApolloF/F Lig4-/-, and Trf2F/FRosa26Cre-ERT1 MEFs were from Wu et al, Molecular Cell, 2010 and Lottersberger et al, Cell, 2015. 293T/17 [HEK 293T/17] (CRL-11268) and Phoenix ECO cells (CRL-3214) were from ATCC, Rockville, MD. RPE-1 cells were from Hegerat et al, EMBO J, 2020. |
| Authentication | RPE-1 cells were validated by whole genome sequencing. MEFs used in the study were automatically genotyped after isolation by TransnetYX for the presence of Apollo or TRF2 "Flox" alleles and/or Ligase IV deletion or RsCre. No cell line validation was performed for 293T or Phoenix ECO cells after acquisition. |
| Mycoplasma contamination | All cell lines in this study routinely tested negative for mycoplasma contamination. |
| Commonly misidentified lines<br>(See ICLAC register) | No cell lines on the current ICLAC register of misidentified cell lines (version 13) were used in this study. |

# Plants

Seed stocks

*Report on the source of all seed stocks or other plant material used. If applicable, state the seed stock centre and catalogue number. If plant specimens were collected from the field, describe the collection location, date and sampling procedures.*

Novel plant genotypes

*Describe the methods by which all novel plant genotypes were produced. This includes those generated by transgenic approaches, gene editing, chemical/radiation-based mutagenesis and hybridization. For transgenic lines, describe the transformation method, the number of independent lines analyzed and the generation upon which experiments were performed. For gene-edited lines, describe the editor used, the endogenous sequence targeted for editing, the targeting guide RNA sequence (if applicable) and how the editor was applied.*

Authentication

*Describe any authentication procedures for each seed stock used or novel genotype generated. Describe any experiments used to assess the effect of a mutation and, where applicable, how potential secondary effects (e.g. second site T-DNA insertions, mosiacism, off-target gene editing) were examined.*

