## [Peer Review File · Nature]

Chromosome end protection by RAP1-mediated inhibition of DNA-PK

Corresponding Author: Dr Max Douglas and Dr Francisca Lottersberger

Version 1:

Reviewer comments:

Referee #1

(Remarks to the Author)

This manuscript focuses on the mechanism of suppression of classical non-homologous end joining (cNHEJ) at telomeres. The authors describe a novel role for the shelterin component, Rap1, in suppressing DNA-PK-mediated cNHEJ. They show that in Apollo-null cells, depletion of Rap1 triggers telomere fusions. To delineate the mechanism by which RAP1 suppresses fusions at telomeres, the authors employ elegant biochemical and structural approaches. They identify a selective DNase I footprinting signature for RAP1 on telomeric sequences bound by DNA-PK. Moreover, the authors show that when recruited to telomeric DNA, RAP1 interacts with KU through its BRCT domains to prevent LIG4 recruitment. Based on these findings, the authors propose a model for the modulation of cNHEJ at telomeres. They suggest that TRF2-mediated end-protection involves a dual mechanism: one acting on telomeres that have been processed by the Apollo nuclease and one involving RAP1 to address telomeres that have not been processed by Apollo. This data underscores the complexity of telomere protection and makes a significant contribution to the field.

Despite these strengths, a major weakness of this manuscript is the lack of data supporting the physiological relevance of the RAP1-mediated suppression of cNHEJ. Rap1 is not an essential gene in mice as well as in human cells. Indeed, depletion of Rap1 does not trigger telomere fusions. This casts doubt on the significant role of Rap1 in telomere protection under physiological conditions. The authors suggest a potential role in senescent cells, based on published data (Lototska et al.), and a potential role in cells that utilise the alternative lengthening of telomeres (ALT) pathways. These speculations are intriguing, yet they are not substantiated by any data in this manuscript. A more detailed discussion and, ideally, additional data would elucidate the biological significance and provide a firmer foundation for the proposed model.

Specific concerns:

1- Physiological Role of RAP1-mediated Suppression of cNHEJ: The manuscript presents compelling biochemical evidence but falls short in demonstrating its significance within cellular contexts, particularly in non-APOLLO-depleted cells. Bridging this gap with additional data would substantiate the importance of RAP1's role in telomere stability and cNHEJ suppression. Furthermore, in Apollo-null cells, the level of fusion is lower than anticipated. Indeed, 50% of telomeres in the absence of Apollo should depend on RAP1-mediated protection. How do the authors interpret this discrepancy?

2- Functional Data Supporting Biochemical Observations: Whilst the manuscript excels in biochemical characterisation, it lacks functional validation of several key findings. For instance, in Fig. 3, the authors demonstrate that RAP1's BRCT domain is crucial for its interaction with DNA-PK. However, the authors do not provide any evidence that this domain plays a role in preventing telomere fusions. A similar argument can be made with the elegant data showing that TRF2's primary function is to recruit RAP1 to telomeres (Fig. 3d). Incorporating cellular assays to complement the biochemical data throughout the manuscript would strengthen the claims and enhance the manuscript's impact. This should be quite straightforward for the authors since they have generated several constructs that could be employed in cells to bolster the biochemical/structural data.

3- Clarification and Additional Controls in Experimental Figures:

Fig. 1c,d: It is unclear how the authors quantified and displayed these data. The figure legend states "Quantification of chromosomes involved in chromatid (c) and chromosome (d) fusions per metaphases...", while the y-axis is labelled "% chr in chromatid/chromosome fusions". It is unclear to me what these values represent.

Fig. 3d: A control using Teb1 or fused to an unrelated factor is required. The protease control is commendable but not sufficient, in my opinion (cleavage could destabilise the proteins).

Fig. 4e, lane 2: This data contradicts the results shown in Fig. 2e. Here, it appears that RAP1 does not require TRF2 to affect DNase I footprinting.

Referee #2

(Remarks to the Author)

DNA-PK, which typically provides a scaffold for Lig4 to initiate non-homologous end joining (NHEJ) at DNA breaks, has been found to bind to telomeres without initiating NHEJ. In this study, the Douglas group demonstrates that the selective inhibition of the catalytic subunit of DNA-PK at telomeres is mediated by components of the Shelterin complex, namely TRF2 and RAP1. These proteins interact with DNA-PK, forming a complex that hinders its end-joining activity at telomeres. Through a combination of biochemical and structural studies, the authors reveal that this inhibition is facilitated by a network of interactions among TRF2, RAP1, KU, and DNA. This network obstructs the recruitment of LIG4 by DNA-PK. Furthermore, they find that inhibition of RAP1, in the absence of APOLLO, induces NHEJ at telomeres.

This study is well-conducted and based on solid biochemical and structural data. It presents novel findings and addresses a significant outstanding question within the field. Moreover, it offers a potential resolution to the paradox surrounding the protective role of RAP1 at telomeres. The manuscript is well-written, the data presentation is clear, and the experimental controls are robust.

One concern relates to the disconnect between Figure 1 and the subsequent sections of the manuscript. Figure 1 features cellular data indicating that loss of RAP1 leads to telomere fusions via NHEJ in the absence of Apollo at a rate of 20%. The focus then shifts in the remainder of the paper to RAP1 and DNA-PK without examining the influence of Apollo and its end resection activity on the joining process. The authors should at least investigate whether an overhang influences the binding affinity of DNA-PK-RAP1. Importantly, they should also correlate the *in vitro* findings with cellular assays, examining the impact of identified mutations in Figure 4 on telomere fusion rates.

An additional point of concern is the magnitude of the effect. TRF2 depletion results in approximately 80% chromosome fusion, whereas Figure 1 reveals that in Apollo-deficient cells treated with sgRAP1, there is a variable frequency of telomere fusion averaging 20%. This raises the question of what factors contribute to the disparity in outcomes between TRF2 loss and RAP1 Apollo DKO conditions. Is it possible that the iDDR motif in TRF2 (PMID: 23389450) independently represses end joining? Alternatively, TRF2 t-loop mutations Ser365 mutation could also be involved PMID: 31723267. A critical experiment would be to complement TRF2 Apollo DKO cells with a TRF2 allele carrying either of these mutations with an RBM mutant.

Last, it would be nice for the authors to speculate in the discussion section the advantages/benefits of having a complex two-tiered protection system.

Referee #3

(Remarks to the Author)

Telomeres are the natural ends of DNA and are required to be protected and maintained in a linear state for genome integrity and cell survival. The authors elucidate a novel RAP1 role in inhibiting telomere fusions by DNA-PK. The authors provide strong evidence for direct interaction of RAP1's BRCT domain with KU as seen from structural and *in vitro* studies on RAP1:DNA-PK complex. The proposed model centers around competitive binding of RAP1 BRCT domain and LIG4 BRCT domain for DNA-PK. Once established, RAP1:DNA-PK interaction inhibits recruitment of LIG4, a crucial factor of c-NHEJ progression thereby negating telomeres end-end fusions.

The novel findings here provide the likely basis of earlier observations of NHEJ being silenced despite the presence of DNA-PK at the telomeres. While this is an important piece of work, additional key experiments are needed to support their claims, and we believe these experiments are within the technical expertise of the authors. Please see our comments/concerns below for details.

Major concerns:

1. It is interesting that the authors don't see any telomeric DNA protection with the addition of shelterin or TRF2 or TRF2-RAP1. This is likely due to the low protein concentration used (5 nM) and the short binding incubation time of 10 min. The previously reported K_d of TRF2-RAP1 to telomeric DNA at comparable salt concentration is ~170 nM (<https://doi.org/10.1093/nar/gkv097>). In Fig. S1d, the authors performed TRF2 EMSA, and at 10 nM TRF2 concentration, most of the DNA is not bound by TRF2. Have the authors tested higher concentrations of shelterin proteins and complexes? And how does it affect the DNase I footprinting assays with DNA-PK? The authors need to test the impact of shelterin proteins and complexes on their DNA-PK DNase I footprinting assays using conditions that provide stoichiometric DNA- and protein-binding.

2. Following the above comment, the DNase I footprinting assays used 30 nM DNA-PKcs and 50 nM KU70/80. Given the cryo-EM model has a 1:1:1 RAP1-KU70/80-DNA-PKcs stoichiometry, shouldn't stoichiometric shelterin proteins (TRF2, RAP1, and TRF2-RAP1 in this case) be used? The authors need to establish the DNase I footprinting outcome using

stoichiometric shelterin proteins. Further, using higher protein concentrations such as 250 nM used in their cryo-EM grids will provide a more relevant connection between their biochemical data and cryo-EM structure.

3. Since TPP1-POT1 are likely located near the telomere ends, it will be important (and possibly more physiologically relevant) to test how a higher concentration of the shelterin complex, such as the TRF2-RAP1-TIN2-TPP1-POT1 complex, would affect the DNase I footprinting pattern with DNA-PK, in comparison with TRF2-RAP1 (see comments 1 and 2). A construct with overhang?

4. Fig. 1a lacks a proof for Apollo knockdown. Also, please show a loading control.

5. The authors struggle to see any visible KU:RAP1 interaction. We wonder how the experiment in Fig. 3e would look in the presence of TRF2 truncation containing only RBM and myb domain or using Teb1:RAP1. Would this strengthen the interaction?

6. DNA-binding assays can complement the authors' DNase I footprint interpretation and are a straightforward assay well within the authors' expertise. Hence, we wonder if they did binding assays and observed complex formation for TRF2/RAP1-DNA-PK-DNA (i.e., a supershift complex)?

7. Please provide input lanes with appropriate loading control for Fig. 5 pulldowns.

8. The authors identified mutants of RAP1 that are inefficient in blocking recruitment of XRCC4/LIG4; however, have the authors confirmed that the mutants show telomere fusions in cells?

9. As RAP1's BRCT domain competes with LIG4's BRCT domain for the KU binding site could the authors comment on relative abundance of LIG4 and RAP1 at telomeres and whether increasing concentrations of LIG4 could weaken RAP1:DNAPK binding?

10. Is it TRF2-RAP1 that recruits DNA-PK or the other way around? Perhaps the authors can comment on this. Also, how would their findings and interpretation integrate with the recent discovery that POT1 caps the telomere C-strand 5' end (10.1126/science.adi2436)? Is POT1 present at telomeres with DNA-PK?

Minor comments:

1. The Fig. 1b legend mentions white and yellow arrows. The figure panel shows green arrow instead of yellow. Please change the writing in figure legend.

2. Please provide domain representation of all the proteins used for structure determination with possibly TRF2 grayed out. This will help the viewers understand what truncations and domains were used for solving the structure.

3. On page 5, line 159 the authors claim the RAP1 binds KU tightly; however, the current body of work suggests the interaction is weak. Fig. 3e shows the interaction is weak even in presence of TRF2/DNA and requires crosslinkers to be detected. This statement is erroneous and lacks proof for the presence of tight binding between KU and RAP1. I suggest the authors rewrite the sentence.

4. For Fig. 2a, please mention both in figure panel next to the cartoon and legend the composition of shelterin complex.

5. Is RAP1 R133 a conserved residue?

6. Fig. 4e, lane 2 shows additional 10-bp protection with only DNA-PK and WT RAP1 but no TRF2. This contradicts the results in Fig. 2e. Perhaps this is a misprint, and should be + for TRF2 instead?

7. In Fig. S1b, *L330R is highlighted. Please mention its significance in the legend.

8. Fig. S1d: Why is the concentration titration for WT TRF2 inconsistent with the mutants? Is that a misprint?

9. Fig. S1e: TEL label is not placed above the indicated lane.

10. Please provide the purification of shelterin complex or provide the reference to this in the Methods section.

11. Please show conservation of residues on KU involved in binding RAP1 BRCT residues. LIG4 BRCT and RAP1 BRCT appear to interact with KU very differently. We wonder if the same residues on KU are involved in binding RAP1 and LIG4.

Version 2:

Reviewer comments:

Referee #1

(Remarks to the Author)

The authors have done an outstanding job in revising the manuscript. The adjustments made not only thoroughly address my previous concerns but also enhance the clarity and depth of the study significantly. The inclusion of additional data has greatly fortified the scientific arguments presented. This meticulous approach has substantially enhanced the manuscript, significantly contributing to our understanding of telomere protection.

Referee #2

(Remarks to the Author)

The authors addressed my concerns. I am in favor of publication.

Referee #3

(Remarks to the Author)

The authors have addressed most, if not all, of my major concerns. Overall, the authors did an excellent job of providing new data to support their interpretations and conclusions.

We have some minor suggestions (missing controls and cryo-EM modeling validation) that we think will further improve the manuscript and solidify their conclusions before its publication:

1. Page 6, line 186: "RAP1 is the most highly conserved component of ..." should be "...most conserved.." and this information is missing reference(s).
2. Fig. 3e: This assay/section of results lacks the addition of TRF2. The authors did have the data from their response to our comment #5. We think that response (and data) should be included in the manuscript as it will strengthen their conclusions – we did not find this new data or discussion in the main text (apologies if we missed finding it).
3. The footprinting data are of high-quality, but we realized that most of the panels lack annotations to the described protein binding sites. Annotating the binding sites (along the vertical gel column) would help readers, who are not familiar with footprinting assays, to follow the discussion in the main text.
4. For figure panels such as S2c, it will be important to include controls such as shelterin/TRF2/RAP1 alone controls (at the tested "high" concentrations). It is interesting that the only control is of the DNAPK alone.
5. Starting Sept 23, all deposited cryo-EM structures have a Q-score metric for each modeled and refined chain. It will be important that the authors provide the average Q-scores for each refined model in their respective column in Table 3. Showing a supplementary figure depicting representative regions of built models with their corresponding map density will be useful for readers to assess the quality of model and structure-function analysis.

We thank the reviewers for taking the time to read and provide insightful feedback on our manuscript. We have responded to their comments by revising the text and including new data into the main and supplementary figures to support and extend our conclusions. We feel this has significantly improved the manuscript. Please find our point-by-point response to each comment below.

Referee 1.

To delineate the mechanism by which RAP1 suppresses fusions at telomeres, the authors employ elegant biochemical and structural approaches. They identify a selective DNase I footprinting signature for RAP1 on telomeric sequences bound by DNA-PK. Moreover, the authors show that when recruited to telomeric DNA, RAP1 interacts with KU through its BRCT domains to prevent LIG4 recruitment. Based on these findings, the authors propose a model for the modulation of cNHEJ at telomeres. They suggest that TRF2-mediated end-protection involves a dual mechanism: one acting on telomeres that have been processed by the Apollo nuclease and one involving RAP1 to address telomeres that have not been processed by Apollo. This data underscores the complexity of telomere protection and makes a significant contribution to the field.

We are glad the reviewer thinks our work makes a significant contribution, and we thank them for their considered comments on the manuscript. We are pleased to provide the responses below, which have validated our model and extended our insights.

A major weakness of this manuscript is the lack of data supporting the physiological relevance of the RAP1-mediated suppression of cNHEJ. Rap1 is not an essential gene in mice as well as in human cells. Indeed, depletion of Rap1 does not trigger telomere fusions. This casts doubt on the significant role of Rap1 in telomere protection under physiological conditions. The authors suggest a potential role in senescent cells, based on published data (Lototska et al.), and a potential role in cells that utilise the alternative lengthening of telomeres (ALT) pathways. These speculations are intriguing, yet they are not substantiated by any data in this manuscript. A more detailed discussion and, ideally, additional data would elucidate the biological significance and provide a firmer foundation for the proposed model.

Thank you for the comment. Indeed, RAP1 is not essential, but we note that the telomeric function of Apollo is also not required for viability or normal growth rates in MEFs^{1,2} suggesting overhang dependent end protection is similarly dispensable. Our finding that cNHEJ is activated in the absence of Apollo when RAP1 is deleted and vice versa explains this non-essentiality by showing that two distinct but equally effective pathways are in place to block end joining at telomeres: overhang dependent end protection and direct inhibition of the NHEJ function of DNA-PK by RAP1. We feel this is an important advance in our understanding of how individual chromosomes are maintained in mammalian cells and demonstrates that the protection from cNHEJ parallels the telomeric blocks to other DNA repair pathways, which also involve multiple mechanisms³.

We agree that our first submission did not clearly consider the biological significance of the RAP1 pathway. To address the concern, we have revised our discussion section with more details provided in response to point 1 below.

Specific concerns:

1- Physiological Role of RAP1-mediated Suppression of cNHEJ: The manuscript presents compelling biochemical evidence but falls short in demonstrating its significance within cellular contexts, particularly in non-APOLLO-depleted cells. Bridging this gap with additional data would substantiate the importance of RAP1's role in telomere stability and cNHEJ suppression.

Given the crucial importance of maintaining monocentric chromosomes, we consider that ensuring robust end protection by providing an inhibitory mechanism to work in parallel to 3' overhangs may alone explain the physiological role of RAP1 in vertebrate cells. RAP1 will buffer instances where Apollo is not yet active and/or 100 % efficient. This is likely to be particularly important given that DNA-PK at blunt leading strand telomeres is required to control Apollo-mediated resection⁴ – if resection fails without RAP1 in place, DNA-PK will be optimally positioned to promote telomere fusions. As noted above, other repair processes are also blocked by multiple telomeric mechanisms³, suggesting such redundancy may in fact be integral to effectively protecting telomeres from DNA repair.

The dependence of Apollo on DNA-PK to resect telomeres⁴ raises the question of how such a mechanism could have evolved without triggering chromosome fusions. Our revised manuscript includes an alphaFold analysis of the RAP1 BRCT:KU interaction that suggests it predates the recruitment of Apollo to telomeres through TRF2. We propose the inhibition of DNA-PK by RAP1 enabled KU and DNA-PKcs at blunt leading strand telomeres to be coopted into a resection role without triggering cNHEJ.

As described in our discussion, it is possible that RAP1 alone blocks cNHEJ in certain contexts. We agree that examining these contexts is important and interesting. However, we hope the reviewer will understand that doing so here would require an extensive set of new experiments, which we feel would go beyond the scope of our manuscript.

Furthermore, in Apollo-null cells, the level of fusion is lower than anticipated. Indeed, 50% of telomeres in the absence of Apollo should depend on RAP1-mediated protection. How do the authors interpret this discrepancy?

We would expect a fusion rate of 50 % if all leading strand telomeres are joined in the absence of Apollo and RAP1. However, the probability that two particles meet in a given time and volume is proportional to their number. Therefore, we consider that not only will 50 % percent of telomeres be available as a substrate for cNHEJ but also not all these ends will necessarily have the chance to fuse, giving rise to a fusion level less than 50 %.

As wild type TRF2 is still present in these cells, we have also considered whether other end protection pathways might be at play. To examine this point, we complemented TRF2^{-/-} MEFs with TRF2 containing the F120A and RBM Δ mutations to prevent Apollo

and RAP1 recruitment, leading to telomere fusions (see figure below). We included further mutations to disrupt either the iDDR⁵ or reduce the level of t-loops through the S367A mutation⁶. The data shows neither mutation increases the level of fusions, suggesting neither t-loops nor the iDDR are restricting cNHEJ here. In fact, deleting the iDDR suppressed fusions. This finding, which is included in our revised manuscript in figures S1b-d, is consistent with previous work showing iDDR deletion enables MRN to restore protective telomeric overhangs in the absence of Apollo².

a. Immunoblot of SV40LT-immortalized TRF2^{F/F} RsCre-ERT1 MEFs transduced with an empty vector control (-), mouse TRF2 or the TRF2 mutants indicated **b**. Representative FISH of metaphase cells expressing the TRF2 mutants indicated 120 h after deletion of endogenous TRF2 with 4-OHT. Telomeres detected with Cy3-(TTAGGG)₃ (green). DNA stained with DAPI (magenta). White and green arrows highlight chromatid- and chromosome fusions respectively. **c.** Percentage of telomeres involved in chromatid and chromosome fusions per metaphases after TRF2 deletion in MEFs expressing the indicated TRF2 mutants. Data from 3 independent experiments, 10 metaphases per experiment (n = 30 total), with median. Statistics by ordinary One-way ANOVA. **d.** Immunoblot of SV40LT-immortalized TRF2^{F/F} RsCre-ERT1 MEFs transduced with an empty vector control (-), mouse TRF2 or the TRF2 mutants indicated **e**. Representative FISH of fragile telomeres detected with Cy3-(TTAGGG)₃ (green) and highlighted in orange. DNA stained with DAPI (magenta). **f.** Representative FISH of metaphase cells expressing the TRF2 mutants indicated 120 h after deletion of endogenous TRF2 with 4-OHT. Telomeres detected with Cy3-(TTAGGG)₃ (green). DNA stained with DAPI (magenta). White and green arrows highlight chromatid- and chromosome fusions respectively. **g.** Percentage telomeres involved in chromatid and chromosome fusions per metaphases after TRF2 deletion in MEFs expressing the indicated TRF2 mutants. Data from 3 independent experiments, 12 metaphases per experiment (n = 36 total), with median. Statistics by ordinary One-way ANOVA.

Lastly, we have examined whether a similar fusion level is observed in human cells. New figures 1d, 1e and S1e-h show that in Apollo/RAP1 double knockout p53^{-/-} RPE-1 cells, 20-30 % of telomeres form fusions that are sensitive to inhibition of DNAPK.

This rate, which is higher than in MEFs, suggests that 40-60 % of leading strand telomeres fuse without Apollo and RAP1. It is possible that the remaining leading strand telomeres are not competent for cNHEJ e.g. because polymerase epsilon terminates before reaching the template end. However, as discussed above, it is also possible that they have not fused because they have not yet encountered another leading strand telomere.

2- Functional Data Supporting Biochemical Observations: Whilst the manuscript excels in biochemical characterisation, it lacks functional validation of several key findings. For instance, in Fig. 3, the authors demonstrate that RAP1's BRCT domain is crucial for its interaction with DNA-PK. However, the authors do not provide any evidence that this domain plays a role in preventing telomere fusions. A similar argument can be made with the elegant data showing that TRF2's primary function is to recruit RAP1 to telomeres (Fig. 3d). Incorporating cellular assays to complement the biochemical data throughout the manuscript would strengthen the claims and enhance the manuscript's impact. This should be quite straightforward for the authors since they have generated several constructs that could be employed in cells to bolster the biochemical/structural data.

We thank the referee for this suggestion. We have performed several new experiments to address this point. Figure 5e shows that unlike wildtype RAP1, the KR/DE and R130E mutants (equivalent to R133E in human RAP1), which are each defective in blocking LIG4 recruitment (new Figs. 5c and d) do not complement the telomere fusions observed in RAP1/Apollo DKO MEFs. We have also made knock in RAP1 mutations in human p53^{-/-} RPE-1 cells, finding that the same KR/DE mutations in the BRCT domain cause telomere fusions in the absence of Apollo (Fig. 5f). This data is consistent with our model that in mouse and human cells, the ability of RAP1 to suppress cNHEJ at telomeres depends on its ability to prevent ligase 4 recruitment.

For the data regarding the primary role of TRF2; we were not suggesting that this is the primary role of TRF2 per se, but rather that the primary role of TRF2 in forming the complex with DNAPK is to recruit RAP1. We have modified the text on page 3, line 97 to make this clear.

3- Clarification and Additional Controls in Experimental Figures:

Fig. 1c,d: It is unclear how the authors quantified and displayed these data. The figure legend states "Quantification of chromosomes involved in chromatid (c) and chromosome (d) fusions per metaphases...", while the y-axis is labelled "% chr in chromatid/chromosome fusions". It is unclear to me what these values represent.

We apologise for the lack of clarity. The graphs reported the percentage of chromosomes that show a chromosome type telomere fusion, or the percentage of chromosomes that show a chromatid type fusion. To make this more consistent, we now show all telomere fusion data from mouse and human cells as '% telomeres fused'.

Fig. 3d: A control using Teb1 or fused to an unrelated factor is required. The protease control is commendable but not sufficient, in my opinion (cleavage could destabilise the proteins).

We see the point and have repeated the experiment with the Teb1 only control as requested. Consistent with our previous figure, Teb1 alone shows no protection of the 10 base pairs next to DNAPK that are footprinted by RAP1 (new Fig. 3d compare lanes 1-4). Our new experiment resolves some additional protection next to the 10 bp footprint when Teb1 and RAP1 are fused together, which is likely to derive from Teb1 preferentially binding DNA close to DNAPK when fused to RAP1.

Fig. 4e, lane 2: This data contradicts the results shown in Fig. 2e. Here, it appears that RAP1 does not require TRF2 to affect DNase I footprinting.

We apologise – this was a typo that has now been corrected. The extra footprint next to DNAPK is entirely TRF2-dependent. Many thanks for picking this up.

Referee 2.

This study is well-conducted and based on solid biochemical and structural data. It presents novel findings and addresses a significant outstanding question within the field. Moreover, it offers a potential resolution to the paradox surrounding the protective role of RAP1 at telomeres. The manuscript is well-written, the data presentation is clear, and the experimental controls are robust.

We thank the referee for their positive comments and input on our manuscript. We are delighted they feel it addresses a significant outstanding question and helps to resolve the telomeric role of RAP1, and we are pleased to provide a point-by-point reply below.

One concern relates to the disconnect between Figure 1 and the subsequent sections of the manuscript. Figure 1 features cellular data indicating that loss of RAP1 leads to telomere fusions via NHEJ in the absence of Apollo at a rate of 20%. The focus then shifts in the remainder of the paper to RAP1 and DNA-PK without examining the influence of Apollo and its end resection activity on the joining process. The authors should at least investigate whether an overhang influences the binding affinity of DNA-PK-RAP1.

Thank you for the comment. We felt analysing the impact of Apollo-mediated resection alongside the role of RAP1 would overly broaden the scope of our study, hence our focus on RAP1 here. As suggested, we have now examined the impact of a 3' overhang on our footprinting system. Figure S2c indicates that DNA-PK can assemble efficiently on a telomeric template containing a 15 nt 3' overhang, binding 4 bp closer to the end than on a blunt template (Fig. S2c compare lanes 2 and 6).

As expected, we observed footprints for RAP1 and DNA-PK on a blunt template when shelterin was added at the same time as KU70/80 and DNA-PKcs, (lanes 2 and 3. Note that the DNase I cleavage within the RAP1 footprint derives from a change in sequence register on the template used here compared with in the rest of the manuscript). However, in equivalent reactions containing the overhang template, assembly of DNA-PK was blocked in a POT1/TPP1-dependent manner (lanes 2-4 and 6-8). This data suggests that as soon as a 3' overhang is formed, the POT1/TPP1 complex can prevent cNHEJ by directly occluding the overhang region, so explaining the integrated activity of RAP1 and Apollo.

Interestingly, the RAP1 footprint formed on the overhang template when POT1/TPP1 was omitted, suggesting the complex can still assemble on a resected telomere if DNA-PK is present.

Importantly, they should also correlate the in vitro findings with cellular assays, examining the impact of identified mutations in Figure 4 on telomere fusion rates.

As suggested, we have examined whether the key interactions identified in our biochemical and structural assays are important for RAP1 to prevent telomere fusions in cells. Figures 5e and S8a-c show that unlike the wild type protein, RAP1 KR/DE and RAP1 R130E (equivalent to R133E in human RAP1), which are each unable to block LIG4 recruitment in vitro (Figs. 5c and d) are defective in suppressing telomere fusions in the absence of Apollo.

We have further validated the importance of LIG4 blocking in p53^{-/-} RPE-1 cells, which also show a telomere fusion phenotype in the absence of both Apollo and RAP1 (described in more detail below). Figures 5f and S8d-g show that knock-in RAP1 mutations K39D, R40E and R55E induce telomere fusions in the absence of Apollo. This data is consistent with our model that in mouse and human cells, the ability of RAP1 to suppress cNHEJ at telomeres depends on its ability to prevent recruitment of LIG4.

TRF2 depletion results in approximately 80% chromosome fusion, whereas Figure 1 reveals that in Apollo-deficient cells treated with sgRAP1, there is a variable frequency of telomere fusion averaging 20%. This raises the question of what factors contribute to the disparity in outcomes between TRF2 loss and RAP1 Apollo DKO conditions. Is it possible that the iDDR motif in TRF2 (PMID: 23389450) independently represses end joining? Alternatively, TRF2 t-loop mutations Ser365 mutation could also be involved PMID: 31723267. A critical experiment would be to complement TRF2 apollo DKO cells with a TRF2 allele carrying either of these mutations with an RBM mutant.

We expect a large part of the different fusion frequencies in TRF2 depleted cells and Apollo/RAP1 DKO cells is due to the lagging strand overhang, which should remain intact and able to protect these ends from cNHEJ in the absence of Apollo¹. In this context, we would expect a maximal fusion rate of 50 % if all leading strand telomeres are joined without Apollo and RAP1. However, the probability that two particles meet in a given time and volume is proportional to their number. Therefore, we consider that not only will 50 % percent of telomeres be available as a substrate for cNHEJ but also that not all of these ends will necessarily have the chance to fuse, giving rise to a fusion level less than 50 %.

As suggested, we have tested the impact of the iDDR and TRF2 t-loop mutations by complementing TRF2^{-/-} MEFs with TRF2 containing the F120A and Δ RBM mutations in combination with Δ iDDR or S367A. The figure below shows that telomere fusions in the absence of RAP1 and Apollo at telomeres were not increased by either the S367A or Δ iDDR mutations, suggesting neither t-loops nor the iDDR are restricting cNHEJ here. In fact, deleting the iDDR suppressed fusions. This finding, which is included in our revised manuscript in figures S1b-d, is consistent with previous work showing

iDDR deletion enables MRN to restore protective telomeric overhangs in the absence of Apollo².

a. Immunoblot of SV40LT-immortalized TRF2F/F RsCre-ERT1 MEFs transduced with an empty vector control (-), mouse TRF2 or the TRF2 mutants indicated **b.** Representative FISH of metaphase cells expressing the TRF2 mutants indicated 120 h after deletion of endogenous TRF2 with 4-OHT. Telomeres detected with Cy3-(TTAGGG)3 (green). DNA stained with DAPI (magenta). White and green arrows highlight chromatid- and chromosome fusions respectively. **c.** Percentage of telomeres involved in chromatid and chromosome fusions per metaphases after TRF2 deletion in MEFs expressing the indicated TRF2 mutants. Data from 3 independent experiments, 10 metaphases per experiment (n = 30 total), with median. Statistics by ordinary One-way ANOVA. **d.** Immunoblot of SV40LT-immortalized TRF2F/F RsCre-ERT1 MEFs transduced with an empty vector control (-), mouse TRF2 or the TRF2 mutants indicated **e.** Representative FISH of fragile telomeres detected with Cy3-(TTAGGG)3 (green) and highlighted in orange. DNA stained with DAPI (magenta). **f.** Representative FISH of metaphase cells expressing the TRF2 mutants indicated 120 h after deletion of endogenous TRF2 with 4-OHT. Telomeres detected with Cy3-(TTAGGG)3 (green). DNA stained with DAPI (magenta). White and green arrows highlight chromatid- and chromosome fusions respectively. **g.** Percentage telomeres involved in chromatid and chromosome fusions per metaphases after TRF2 deletion in MEFs expressing the indicated TRF2 mutants. Data from 3 independent experiments, 12 metaphases per experiment (n = 36 total), with median. Statistics by ordinary One-way ANOVA.

To examine the level of telomere fusions further, we tested whether a similar percentage of telomeres fuse in non-transformed human cells. Figures 1d, 1e and S1e-h show that 20-30 % of telomeres fuse in a DNAPK-dependent manner in p53/- RPE-1 cells lacking Apollo and RAP1. This level, which is higher than in MEFs, suggests that on average 40-60 % of leading strand telomeres fuse when Apollo and RAP1 are absent. It is possible that the remaining leading strand telomeres are not competent for cNHEJ for example because polymerase epsilon terminates before reaching the template end. However, as discussed above, it is also possible that they have not fused because they have not encountered another leading strand telomeric end.

Last, it would be nice for the authors to speculate in the discussion section the advantages/benefits of having a complex two-tiered protection system.

Thank you for the suggestion. We have now rewritten our discussion section to include a more detailed consideration of the benefits of having two parallel mechanisms preventing cNHEJ at telomeres.

We suggest the use of RAP1 and 3' overhangs will provide robust maintenance of individual chromosomes by ensuring that DNA-PK cannot activate cNHEJ prior to resection by Apollo or if this process fails. This is likely to be particularly crucial because DNAPK is required for Apollo-mediated resection⁴ and so will be optimally positioned to promote telomere fusions. Given that other repair pathways are also blocked by multiple mechanisms at chromosome ends³, such redundancy may be integral to effectively protecting telomeres from DNA repair.

The dependence of Apollo-mediated resection on DNAPK⁴ raises the question of how such a mechanism could have evolved without DNAPK triggering cNHEJ. Using alphafold, our revised manuscript demonstrates that the RAP1 BRCT:KU interaction is likely to be widely conserved across metazoa (Fig. S9), and therefore to predate the recruitment of Apollo to telomeres via TRF2⁸. We propose that RAP1-mediated inhibition of end joining enabled KU and DNA-PKcs at leading strand telomeres to be coopted into a resection role in vertebrate species by preventing them from triggering cNHEJ.

Lastly, RAP1 alone has been suggested to protect telomeres from cNHEJ in senescent cells^{9,10}, suggesting there may be contexts in which it is the primary protective factor. We note that RAP1 can block LIG4 recruitment in vitro with only a small number of telomeric repeats suggesting the mechanism we describe may be effective at preventing cNHEJ at critically short telomeres in these cells.

Referee 3.

Telomeres are the natural ends of DNA and are required to be protected and maintained in a linear state for genome integrity and cell survival. The authors elucidate a novel RAP1 role in inhibiting telomere fusions by DNA-PK. The authors provide strong evidence for direct interaction of RAP1's BRCT domain with KU as seen from structural and in vitro studies on RAP1:DNA-PK complex. The proposed model centers around competitive binding of RAP1 BRCT domain and LIG4 BRCT domain for DNA-PK. Once established, RAP1:DNA-PK interaction inhibits recruitment of LIG4, a crucial factor of c-NHEJ progression thereby negating telomeres end-end fusions.

The novel findings here provide the likely basis of earlier observations of NHEJ being silenced despite the presence of DNA-PK at the telomeres. While this is an important piece of work, additional key experiments are needed to support their claims, and we believe these experiments are within the technical expertise of the authors. Please see our comments/concerns below for details.

We thank the referee for their insightful and helpful comments. We are pleased to provide the point-by-point reply below.

Major concerns:

1. *It is interesting that the authors don't see any telomeric DNA protection with the addition of shelterin or TRF2 or TRF2-RAP1. This is likely due to the low protein concentration used (5 nM) and the short binding incubation time of 10 min. The previously reported K_d of TRF2-RAP1 to telomeric DNA at comparable salt concentration is ~170 nM (<https://doi.org/10.1093/nar/gkv097>). In Fig. S1d, the authors performed TRF2 EMSA, and at 10 nM TRF2 concentration, most of the DNA is not bound by TRF2. Have the authors tested higher concentrations of shelterin proteins and complexes? And how does it affect the DNase I footprinting assays with DNA-PK? The authors need to test the impact of shelterin proteins and complexes on their DNA-PK DNase I footprinting assays using conditions that provide stoichiometric DNA- and protein-binding.*

The shelterin and TRF2:RAP1 concentrations used in our assays were optimised to examine specific footprints next to or around DNAPK without general protection of the template. Even with low shelterin concentrations, some general protection of telomeric DNA can be observed by comparing signal intensity in lanes 1 and 6 in Fig. 2b.

To address the point directly, we have tested higher concentrations of shelterin and TRF2:RAP1 in the assay. As shown in figure S2b, with up to 80 nM TRF2:RAP1 or shelterin (where all template molecules are bound by several TRF2:RAP1 complexes - Fig. S2f) we observe the same 10 bp footprint next to DNAPK, but also a more extensive overall protection of the template, which is likely to be bound by many shelterin molecules under these conditions. Please also see points 2 and 3 below.

2. *Following the above comment, the DNase I footprinting assays used 30 nM DNA-PKcs and 50 nM KU70/80. Given the cryo-EM model has a 1:1:1 RAP1-KU70/80-DNA-PKcs stoichiometry, shouldn't stoichiometric shelterin proteins (TRF2, RAP1, and TRF2-RAP1 in this case) be used? The authors need to establish the DNase I footprinting outcome using stoichiometric shelterin proteins. Further, using higher protein concentrations such as 250 nM used in their cryo-EM grids will provide a more relevant connection between their biochemical data and cryo-EM structure.*

Concentrations of KU70/80 and DNA-PKcs were optimised at the beginning of the project to generate a robust DNAPK footprint in our biochemical assays, hence the excess used. We have now tested high equimolar protein concentrations as requested. Figure S2b shows that with 20 nM DNA template and 250 nM each of KU, DNA-PKcs and TRF2/RAP1, we observe the same footprinting pattern as with lower, non-stoichiometric protein concentrations. Thus, formation of the TRF2:RAP1:DNAPK complex is apparently robust to changes in the absolute and relative protein concentrations used in the assay.

3. *Since TPP1-POT1 are likely located near the telomere ends, it will be important (and possibly more physiologically relevant) to test how a higher concentration of the shelterin complex, such as the TRF2-RAP1-TIN2-TPP1-POT1 complex, would affect the DNase I footprinting pattern with DNA-PK, in comparison with TRF2-RAP1 (see comments 1 and 2). A construct with overhang?*

The impact of increasing concentrations of either shelterin or TRF2:RAP1 in our footprinting assay is described in response to point 1 above. We have also examined the impact of a 3' overhang as suggested. Figure S2c indicates that DNA-PK can assemble efficiently on a template containing a 15 nt 3' overhang, binding 4 bp closer to the end than on a blunt template (Fig. S2c compare lanes 2 and 6).

As expected, we observed footprints for RAP1 and DNA-PK on a blunt template when shelterin (containing TRF1, TRF2, POT1, TPP1, TIN2 and RAP1) was added with KU and DNA-PKcs (lanes 2 and 3. Please note the DNase I cleavage within the RAP1 footprint derives from a change in the sequence register of the template used here compared with in the rest of the manuscript). However, in equivalent reactions containing the overhang template, assembly of DNA-PK was blocked in a POT1/TPP1-dependent manner (lanes 2-4 and 6-8). This data suggests that as soon as a 3' overhang is formed, the POT1/TPP1 complex can prevent cNHEJ by directly occluding the overhang region, so explaining the integrated activity of RAP1 and Apollo.

Interestingly, without POT1/TPP1, the RAP1 footprint next to DNA-PK is also observed on the overhang template, suggesting the complex can still form on a resected telomere if DNA-PK is present.

We have also repeated these experiments with higher shelterin concentrations - the figure below shows that with 40 nM shelterin, 100 nM DNA-PK components and 8 nM DNA (to avoid overall protection of the template) the same footprinting patterns as in Fig. S2c are observed.

Figure showing that the same footprinting pattern as in Fig. S2c is observed on blunt and overhang templates with 40 nM shelterin, 100 nM DNA-PK and 8 nM DNA.

4. Fig. 1a lacks a proof for Apollo knockdown. Also, please show a loading control.

There is unfortunately no good antibody for mouse Apollo. However, these cells have been extensively characterised^{1,4,11}, and the LIG4-independent chromatid fusion phenotype we observe (e.g. in Fig. 1b) provides a functional readout that Apollo has been successfully removed. Moreover, we have now confirmed the phenotype in our revised manuscript by complementing TRF2^{-/-} MEFs with TRF2 lacking the binding sites for Apollo and RAP1 (Fig. S1b-d). A loading control is now added as requested – we apologise that this was not included in the original submission. Please also note that loss of human Apollo, for which there is an effective antibody, is confirmed for all our new RPE-1 cell data.

5. The authors struggle to see any visible KU:RAP1 interaction. We wonder how the experiment in Fig. 3e would look in the presence of TRF2 truncation containing only RBM and myb domain or using *Teb1:RAP1*. Would this strengthen the interaction?

Thank you for this idea. Unfortunately, the *Teb1* DBD makes crosslinks in the absence of RAP1, but we have tested whether recruitment of RAP1 to the template is limiting for crosslinking here by adding the TRF2 truncation suggested.

As shown in the figure below, adding the TRF2 truncation led to an upshift in the size of the crosslinked species, consistent with this fragment of TRF2 binding to RAP1 and having been incorporated into the complex with KU. However, we did not observe more efficient crosslinking under these conditions (compare intensity of crosslinked band in FLAG Western). Given the variety of parameters that can feed into crosslinking efficiency, we favour that it is limited by other factors such as the orientation of crosslinkable lysines, or the kinetics of the interaction rather than just the proximity of RAP1 to KU.

Crosslinking was performed as in figure 3e, except a fragment of TRF2 (316-542) was added at 200 nM alongside KU and RAP1 as indicated. Right hand panel shows TRF2 was incorporated into the KU/RAP1 complex, but the level of KU in the crosslinked species does not increase (left hand panel). The increase in strep signal may derive from some residual strep-tag on the TRF2 fragment.

6. DNA-binding assays can complement the authors' DNase I footprint interpretation and are a straightforward assay well within the authors' expertise. Hence, we wonder if they did binding assays and observed complex formation for TRF2/RAP1-DNA-PK-DNA (i.e., a supershift complex)?

As the components studied here each independently interact with DNA, we have not tried these kind of assays - we considered we would be unable to distinguish between a supershifted band generated by multiple proteins binding independently to the same DNA molecule, and a supershifted band generated by proteins interacting with one-another on DNA as a complex. These limitations were in fact the original motivation for pursuing the DNase I footprinting approach, which allows us to distinguish between these possibilities by spatially resolving binding sites along the template.

7. Please provide input lanes with appropriate loading control for Fig. 5 pulldowns.

We apologise for not having included this in the original submission. All pulldown figures in the manuscript (Figs. 5 and S7) are now shown with inputs.

8. The authors identified mutants of RAP1 that are inefficient in blocking recruitment of XRCC4/LIG4; however, have the authors confirmed that the mutants show telomere fusions in cells?

We thank the referee for raising this central point. We have now examined whether key interactions identified in our biochemical assays are important for RAP1 to prevent telomere fusions in cells. Figures 5e and S8a-c show that unlike the wildtype protein, RAP1 KR/DE and RAP1 R130E (equivalent to R133E on human RAP1), which are each unable to block LIG4 recruitment in vitro (Figs. 5c and d) are defective in suppressing telomere fusions in the absence of Apollo.

We have further validated the importance of LIG4 blocking in human p53^{-/-} RPE-1 cells, which also exhibit DNAPK-dependent telomere fusions in the absence of Apollo and RAP1 (Fig. 1c and d and Fig. S1e-i). Figures 5f and S8d-g show that knock-in RAP1 mutations K39D, R40E and R55E in RPE-1 cells induce telomere fusions in the absence of Apollo. This data is consistent with our model that in mouse and human cells, the ability of RAP1 to suppress cNHEJ at telomeres depends on its ability to prevent recruitment of XRCC4/LIG4.

9. As RAP1's BRCT domain competes with LIG4's BRCT domain for the KU binding site could the authors comment on relative abundance of LIG4 and RAP1 at telomeres and whether increasing concentrations of LIG4 could weaken RAP1:DNAPK binding?

Certainly. Proteomic studies indicate that LIG4 is in excess over RAP1 in human cells¹². However, there is an estimated one molecule of RAP1 per 50-60 bp of telomeric DNA in HeLa cells, suggesting it is abundant at telomeres¹³. We established the KU pulldown assay in figure 5 with a 6-fold excess of XRCC4/LIG4 over TRF2/RAP1 and have now analysed how the assay responds to even higher XRCC4/LIG4 concentrations. Fig. S7b shows that even a 24-fold excess of XRCC4/LIG4 over TRF2/RAP1 (120 nM vs 5 nM) is blocked from binding to DNAPK. Fig. S7c shows that this is also the case if XRCC4/LIG4 is preincubated with KU, DNA-PKcs and the template prior to the addition of TRF2/RAP1. Thus, TRF2/RAP1 exerts

a robust block to LIG4 recruitment and is unlikely to be significantly weakened by increasing LIG4 concentrations in our opinion.

10. *Is it TRF2-RAP1 that recruits DNA-PK or the other way around? Perhaps the authors can comment on this. Also, how would their findings and interpretation integrate with the recent discovery that POT1 caps the telomere C-strand 5' end (10.1126/science.adi2436)? Is POT1 present at telomeres with DNA-PK?*

We have not observed an effect of TRF2:RAP1 on the efficiency of DNA-PK assembly in either our footprinting assay (please see the figure below), or in the KU pulldown assay in figures 5b and S7. We therefore think it is unlikely that DNA-PK is recruited by TRF2/RAP1. In contrast, figure 3d shows that fusing Teb1 to RAP1 is sufficient to recruit the DNA binding domain to a terminal position (compare lanes 2, 4 and 5). Thus, our working model is that DNA-PK assembled on telomeric DNA either binds TRF2/RAP1 that is already close to the template end or recruits it to a terminal position rather than the other way round.

DNase I footprinting assay showing DNA-PK assembly at different KU and DNA-PK concentrations is unaffected by the presence of TRF2/RAP1.

Regarding POT1, the new data in response to point 3 above suggests that capping by POT1/TPP1 can prevent the assembly of DNAPK on telomeres terminating in an overhang (Fig. S2c). Combined with the proposal that t-loops block DNA-PK assembly on these ends, binding of POT1 and DNA-PK to telomeric ends are likely to be mutually exclusive in our opinion.

Minor comments:

1. The Fig. 1b legend mentions white and yellow arrows. The figure panel shows green arrow instead of yellow. Please change the writing in figure legend.

Thanks for spotting. This has now been corrected.

2. Please provide domain representation of all the proteins used for structure determination with possibly TRF2 grayed out. This will help the viewers understand what truncations and domains were used for solving the structure.

This has been included as requested.

3. On page 5, line 159 the authors claim the RAP1 binds KU tightly; however, the current body of work suggests the interaction is weak. Fig. 3e shows the interaction is weak even in presence of TRF2/DNA and requires crosslinkers to be detected. This statement is erroneous and lacks proof for the presence of tight binding between KU and RAP1. I suggest the authors rewrite the sentence.

This section of the manuscript has been extensively revised with the statement that it binds tightly now removed.

4. For Fig. 2a, please mention both in figure panel next to the cartoon and legend the composition of shelterin complex.

This has been added as requested

5. Is RAP1 R133 a conserved residue?

Yes, R133 is very highly conserved. We realised that the original alignment now in Fig. S5c was misnumbered and also contained two incorrect sequences. Both errors have now been corrected.

6. Fig. 4e, lane 2 shows additional 10-bp protection with only DNA-PK and WT RAP1 but no TRF2. This contradicts the results in Fig. 2e. Perhaps this is a misprint, and should be + for TRF2 instead?

Thank you for spotting. This is indeed a misprint and has now been corrected.

7. In Fig. S1b, *L330R is highlighted. Please mention its significance in the legend.

This mutation corresponds to L288R in the shorter 500 amino acid TRF2 isoform and blocks RAP1 binding¹⁴. This has been included into the legend for Fig. S2d as requested.

8. Fig. S1d: Why is the concentration titration for WT TRF2 inconsistent with the mutants? Is that a misprint

Indeed, this was a misprint and has now been corrected (now in figure S2f) – the same concentration range was tested for each version of TRF2.

9. Fig. S1e: TEL label is not placed above the indicated lane.

Thanks for spotting – this has now been corrected.

10. Please provide the purification of shelterin complex or provide the reference to this in the Methods section.

Please find the shelterin purification protocol in the methods section 'TRF2, RAP1 and shelterin purification', now on page 9.

11. Please show conservation of residues on KU involved in binding RAP1 BRCT residues. LIG4 BRCT and RAP1 BRCT appear to interact with KU very differently. We wonder if the same residues on KU are involved in binding RAP1 and LIG4.

Indeed, RAP1 and LIG4 do bind to KU very differently but using the same KU residues. This is demonstrated directly by Fig. S7a, which shows that the same three charge reversal mutations on KU70 and KU80 that prevent the RAP1 footprint (Fig. 4g) also prevent recruitment of LIG4 in our pulldown assay (compare lanes 7 and 8). Figure S6 now includes an alignment of the relevant sections of KU70 and KU80 as requested, with KU70 D496 and E499 and KU80 D327 (mutated in the KU DE/KR mutant) highlighted. Our new alphafold analysis in figure S9 suggests that KU across a wide range of metazoan species is likely to bind the RAP1 BRCT domain at this position.

References

- 1 Wu, P., Takai, H. & de Lange, T. Telomeric 3' overhangs derive from resection by Exo1 and Apollo and fill-in by POT1b-associated CST. *Cell* **150**, 39-52 (2012). <https://doi.org/10.1016/j.cell.2012.05.026>
- 2 Myler, L. R. *et al.* DNA-PK and the TRF2 iDDR inhibit MRN-initiated resection at leading-end telomeres. *Nat Struct Mol Biol* **30**, 1346-1356 (2023). <https://doi.org/10.1038/s41594-023-01072-x>
- 3 Sfeir, A. & de Lange, T. Removal of shelterin reveals the telomere end-protection problem. *Science* **336**, 593-597 (2012). <https://doi.org/10.1126/science.1218498>
- 4 Sonmez, C. *et al.* DNA-PK controls Apollo's access to leading-end telomeres. *Nucleic Acids Res* **52**, 4313-4327 (2024). <https://doi.org/10.1093/nar/gkae105>
- 5 Okamoto, K. *et al.* A two-step mechanism for TRF2-mediated chromosome-end protection. *Nature* **494**, 502-505 (2013). <https://doi.org/10.1038/nature11873>
- 6 Sarek, G. *et al.* CDK phosphorylation of TRF2 controls t-loop dynamics during the cell cycle. *Nature* **575**, 523-527 (2019). <https://doi.org/10.1038/s41586-019-1744-8>
- 7 Hockemeyer, D., Daniels, J. P., Takai, H. & de Lange, T. Recent expansion of the telomeric complex in rodents: Two distinct POT1 proteins protect mouse telomeres. *Cell* **126**, 63-77 (2006). <https://doi.org/10.1016/j.cell.2006.04.044>
- 8 Myler, L. R. *et al.* The evolution of metazoan shelterin. *Genes Dev* **35**, 1625-1641 (2021). <https://doi.org/10.1101/gad.348835.121>
- 9 Lototska, L. *et al.* Human RAP1 specifically protects telomeres of senescent cells from DNA damage. *EMBO Rep* **21**, e49076 (2020). <https://doi.org/10.15252/embr.201949076>
- 10 Martínez, P., Gómez-López, G., Pisano, D. G., Flores, J. M. & Blasco, M. A. A genetic interaction between RAP1 and telomerase reveals an unanticipated

- role for RAP1 in telomere maintenance. *Aging Cell* **15**, 1113-1125 (2016).
<https://doi.org/10.1111/ace1.12517>
- 11 Wu, P., van Overbeek, M., Rooney, S. & de Lange, T. Apollo contributes to G overhang maintenance and protects leading-end telomeres. *Mol Cell* **39**, 606-617 (2010). <https://doi.org/10.1016/j.molcel.2010.06.031>
- 12 Hein, M. Y. *et al.* A human interactome in three quantitative dimensions organized by stoichiometries and abundances. *Cell* **163**, 712-723 (2015).
<https://doi.org/10.1016/j.cell.2015.09.053>
- 13 Takai, K. K., Hooper, S., Blackwood, S., Gandhi, R. & de Lange, T. In vivo stoichiometry of shelterin components. *J Biol Chem* **285**, 1457-1467 (2010).
<https://doi.org/10.1074/jbc.M109.038026>
- 14 Chen, Y. *et al.* A conserved motif within RAP1 has diversified roles in telomere protection and regulation in different organisms. *Nat Struct Mol Biol* **18**, 213-221 (2011). <https://doi.org/10.1038/nsmb.1974>

Response to referees

Referee #3:

The authors have addressed most, if not all, of my major concerns. Overall, the authors did an excellent job of providing new data to support their interpretations and conclusions.

Many thanks for the positive comments on our revised manuscript.

We have some minor suggestions (missing controls and cryo-EM modeling validation) that we think will further improve the manuscript and solidify their conclusions before its publication:

1. Page 6, line 186: “RAP1 is the most highly conserved component of ...” should be “..most conserved..” and this information is missing reference(s).

This has been changed as requested

2. Fig. 3e: This assay/section of results lacks the addition of TRF2. The authors did have the data from their response to our comment #5. We think that response (and data) should be included in the manuscript as it will strengthen their conclusions – we did not find this new data or discussion in the main text (apologies if we missed finding it).

As requested, this data has now been included as Extended Data figure 2i

3. The footprinting data are of high-quality, but we realized that most of the panels lack annotations to the described protein binding sites. Annotating the binding sites (along the vertical gel column) would help readers, who are not familiar with footprinting assays, to follow the discussion in the main text.

This has been included for all footprinting experiments as requested

4. For figure panels such as S2c, it will be important to include controls such as shelterin/TRF2/RAP1 alone controls (at the tested “high” concentrations). It is interesting that the only control is of the DNAPK alone.

As requested, a ‘shelterin only’ control has now been included alongside the ‘high concentration’ experiment in Extended data figure 2c.

5. Starting Sept 23, all deposited cryo-EM structures have a Q-score metric for each modeled and refined chain. It will be important that the authors provide the average Q-scores for each refined model in their respective column in Table 3. Showing a supplementary figure depicting representative regions of built models with their corresponding map density will be useful for readers to assess the quality of model and structure-function analysis.

The Q-score is now reported in Extended data table 1 for the composite EM map. Several main and Extended data figures such as Fig. 4a-c, Extended Data Fig. 5a, b and g show regions of the built model with their corresponding map density